# Rdh54/Tid1 inhibits Rad51-Rad54-mediated D-loop formation and limits D-loop length

Shanaya Shital Shah[1], Stella Hartono[2], Aurèle Piazza[1,3], Vanessa Som[1], William Wright[1,4], Frédéric Chédin[2], Wolf-Dietrich Heyer[1,2]*

[1]Department of Microbiology and Molecular Genetics, University of California, Davis, Davis, United States; [2]Department of Molecular and Cellular Biology, University of California, Davis, Davis, United States; [3]CR CNRS UMR5239, Team Genome Mechanics, Laboratory of Biology and Modelling of the Cell, Ecole Normale Supérieure de Lyon 46, Lyon, France; [4]Mammoth Biosciences, South San Francisco, United States

**Abstract** Displacement loops (D-loops) are critical intermediates formed during homologous recombination. Rdh54 (a.k.a. Tid1), a Rad54 paralog in *Saccharomyces cerevisiae,* is well-known for its role with Dmc1 recombinase during meiotic recombination. Yet contrary to Dmc1, Rdh54/Tid1 is also present in somatic cells where its function is less understood. While Rdh54/Tid1 enhances the Rad51 DNA strand invasion activity in vitro, it is unclear how it interplays with Rad54. Here, we show that Rdh54/Tid1 inhibits D-loop formation by Rad51 and Rad54 in an ATPase-independent manner. Using a novel D-loop Mapping Assay, we further demonstrate that Rdh54/Tid1 uniquely restricts the length of Rad51-Rad54-mediated D-loops. The alterations in D-loop properties appear to be important for cell survival and mating-type switch in haploid yeast. We propose that Rdh54/Tid1 and Rad54 compete for potential binding sites within the Rad51 filament, where Rdh54/Tid1 acts as a physical roadblock to Rad54 translocation, limiting D-loop formation and D-loop length.

*For correspondence:
wdHeyer@ucdavis.edu

## Introduction

Homologous recombination (HR) is a universal DNA repair pathway that uses an intact homologous donor for the repair of double-stranded DNA breaks (DSBs), stalled or collapsed forks and inter-strand crosslinks (*Kowalczykowski, 2015*; *Wright et al., 2018*). Consequently, defects in HR or its regulation lead to genomic instability, chromosomal aberrations, tumorigenesis and cell death.

HR begins by resection of the broken DNA molecule, followed by recruitment of the Rad51 recombinase to form a filament on the single-stranded DNA (ssDNA). The Rad51 filament then executes homology search and DNA strand invasion into a homologous duplex donor DNA (*Kowalczy-kowski, 2015*). In the resulting pairing intermediate, the Rad54 motor protein displaces Rad51, while threading out a heteroduplex DNA (hDNA) and a displaced strand, to create a stable interme-diate called the displacement loop (or D-loop) (*Wright and Heyer, 2014*). The D-loop thus features a displaced ssDNA, a Rad51-free hDNA and DNA strand exchange junctions at both extremities of the hDNA.

The D-loop is a pivotal intermediate of the HR pathway, acted upon by various types of enzymes. D-loops containing an annealed 3′-OH end can be extended by a DNA polymerase, which commits to the use of the donor as a template for the repair (*Wright et al., 2018*). Helicases and/or topoiso-merases such as Sgs1-Top3-Rmi1, Mph1, and Srs2 revert D-loops (*Fasching et al., 2015*; *Liu et al., 2017*; *Piazza et al., 2019*; *Prakash et al., 2009*; *Putnam et al., 2009*). This reversibility presumably enforces the fidelity of the repair pathway (*Piazza and Heyer, 2019*; *Putnam and Kolodner, 2017*).

The D-loop disruption mechanism is enhanced by mismatch repair proteins at mismatched hDNA in a process termed heteroduplex rejection (*Chakraborty et al., 2016*). Furthermore, the dynamic nature of D-loops endowed by these enzymes also prevents concomitant invasions, either of both broken ends in the same donor molecule leading to double Holliday Junctions (dHJ) (*Wright et al., 2018*; *Piazza and Heyer, 2019*), or of a single end into two different donors leading to multi-invasions (MI) (*Piazza and Heyer, 2018*; *Piazza et al., 2017*), thus inhibiting downstream covalent alterations of the donors mediated by structure-selective endonucleases (SSEs). Indeed, both crossovers and MI-induced rearrangements increase in *mph1*, *sgs1-top3-rmi1*, and *srs2* mutants (*Ira et al., 2003*; *Piazza et al., 2019*; *Piazza et al., 2017*; *Prakash et al., 2009*; *Prakash et al., 2009*).

Structural features of the D-loop are likely cues for the various proteins acting upon them. For instance, D-loops with a 3′ flap instead of an annealed 3′-OH cannot be readily extended, but instead exhibit a loading pad for the aforementioned 3′−5′ helicases, likely promoting their disruption. Second, D-loops exhibiting longer hDNA may be harder to disrupt. Consequently, the DNA strand invasion apparatus (which by definition drives the pathway forward) may already elicit the backward reaction by determining the structure of the D-loop, and thus be part of the regulatory branch of HR promoting genome stability. However, the interplay of factors involved in DNA strand invasion and their consequence on D-loop structure is poorly defined.

Rdh54 (also known as Tid1), is a *Saccharomyces cerevisiae* Rad54 paralog, conserved in eukaryotes, and a member of the SWI2/SNF2 family of helicase-like chromatin-remodelers (*Eisen, 1995*; *Flaus and Owen-Hughes, 2011*). The biochemical properties of Rad54 and Rdh54/Tid1 are exceedingly similar in terms of ATPase activity, translocation on dsDNA (*Nimonkar et al., 2007*; *Bianco et al., 2007*), removal of Rad51 bound to dsDNA (*Holzen et al., 2006*; *Santa Maria et al., 2013*; *Solinger et al., 2002*), stimulation of D-loop reactions by Rad51 and Dmc1 (the meiosis-specific recombinase) (*Nimonkar et al., 2012*; *Wright and Heyer, 2014*), and ability to disrupt joint molecules (*Nimonkar et al., 2007*; *Wright and Heyer, 2014*). Careful biochemical investigations indicated that Rad51 primarily works with Rad54, and Dmc1 with Rdh54/Tid1 (*Nimonkar et al., 2012*). However, and contrary to Dmc1, Rdh54/Tid1 is expressed in mitotically dividing cells (*Lee et al., 2001*), suggesting that it has a unique function during somatic HR.

In somatic cells, Rdh54/Tid1 is phosphorylated in response to DNA damage by Mec1 (*Ferrari et al., 2013*) and is recruited to DSBs in a Rad51-dependent manner (*Kwon et al., 2008*; *Lisby et al., 2004*). Rdh54/Tid1 also interacts with Rad51 (*Santa Maria et al., 2013*) and can promote the DNA strand invasion activity of Rad51 in vitro (*Nimonkar et al., 2012*; *Petukhova et al., 2000*). Yet, deletion of *RHD54/TID1* only subtly affects DSB repair in mitotic cells, unless sister chromatid-based repair is eliminated (*Aguilera and Klein, 1988*; *Arbel et al., 1999*; *Ira and Haber, 2002*; *Klein, 1997*). However, Rdh54/Tid1 negatively affects D-loops in vivo in budding yeast (*Piazza et al., 2019*). Deletion of *RDH54/TID1* results in a marked increase in the D-loop signal by physical detection of nascent D-loops using the D-loop capture (DLC) assay. Due to the limitation of the DLC assay used, it is unclear if the increase in D-loop signal is due to an increase in total D-loop levels, an increase in D-loop length, where longer D-loops may be more likely to be stably cross-linked and detected by the assay, or both (*Piazza et al., 2019*). Moreover, the ATPase-defective *rdh54-KR/tid1-KR* had no change in the D-loop signal compared to the wild-type strain. This suggested a novel ATPase-independent role of Rdh54/Tid1 on somatic D-loops, in contrast to a motor activity dependent downstream role of Rdh54/Tid1 on crossover frequency and DNA repair (*Piazza et al., 2019*). Hence, we decided to examine the biochemical properties of Rdh54/Tid1 in reconstituted in vitro recombination containing also Rad54, as there is no information available on how these two proteins interact during in vitro recombination.

Here, we show that Rdh54/Tid1 inhibits Rad54-mediated D-loop formation in vitro and in vivo. The inhibition is independent of its motor activity, by competing with Rad54 in a concentration-dependent manner. Moreover, to address any potential effect on D-loop length, we developed an in vitro D-loop Mapping Assay (DMA) (*Shah et al., 2020*) to determine D-loop length and position at single-molecule with near base pair resolution. The assay is based on bisulfite sequencing and adapted from mapping R-loops (*Malig et al., 2020*; *Yu et al., 2003*). Using the DMA in vitro, we show that Rdh54/Tid1 also limits D-loop lengths formed by Rad54 (D-loops < 300 nt). These alterations in D-loops by Rdh54/Tid1 are subsequently crucial in maintaining cell viability and kinetics of D-loop extension in a homology-length dependent way. Together these findings highlight an antagonistic relationship between the two Swi2/Snf2 ATPases and their function in HR.

## Results

Hereon, Rdh54 (protein), and *RDH54* (gene) are denoted as Tid1 and *TID1*, respectively, despite Rdh54 and *RDH54* being the Saccharomyces Genome Database recognized nomenclature. Tid1 and *TID1* are used to avoid misreading and confusion with the closely spelled Rad54 protein and *RAD54* gene.

### Tid1 inhibits D-loop formation in vitro

As Rad54 and Tid1 are expressed and recruited to the site of a DSB in somatic cells, we sought to gain insights into their interplay using a reconstituted DNA strand invasion reaction with purified RPA, Rad51, and DNA substrates mimicking physiological resection length (*Figure 1A*). Purified GST-Tid1 or its ATPase-defective mutant GST-Tid1-K318R (from now on referred to as Tid1 and Tid1-KR) (*Figure 1—figure supplement 1A,B*) were titrated into the D-loop reaction, 5 min before the addition of double-stranded donor DNA (dsDNA) and Rad54, but after Rad51 filament formation (*Figure 1A*) (for details, see Materials and methods). We used linear duplex DNA as a donor as to not limit the length of the D-loop by topological constraints. As shown in *Figure 1B*, the presence of Tid1 significantly inhibits D-loop formation by Rad51 and Rad54 in a concentration-dependent manner. There is a fourfold decrease in the D-loop level at the highest (14x) Tid1 concentration (*Figure 1C*, *Figure 1—figure supplement 1C*). The molar ratio of the invading DNA and the duplex donor is 1:1. The D-loop quantifications were made relative to the dsDNA donor, not the substrate (for details, see Materials and methods). Tid1 thus inhibits D-loop formation by Rad54, despite being able to promote DNA strand invasion of Rad51 by itself (*Figure 1—figure supplement 1E*), as previously reported (*Nimonkar et al., 2012*; *Petukhova et al., 2000*). We note that this stimulation of Rad51-mediated DNA strand invasion activity of Tid1 is significantly less efficient than Rad54 (*Figure 1—figure supplement 1E*), similar to prior observations (*Nimonkar et al., 2012*).

Additionally, the ATPase-defective mutant Tid1-KR, lacking the ability to translocate on dsDNA and catalyze D-loop formation (*Nimonkar et al., 2012*; *Chi et al., 2006*), also inhibited D-loop formation (*Figure 1D and E*, *Figure 1—figure supplement 1D*). Tid1-KR, like wild-type Tid1 (Tid1-WT), inhibited D-loops increasingly at higher concentrations, enabling a ninefold decrease in the D-loop levels at the highest Tid1-KR concentration (7x). This inhibition was more efficient than that mediated by the Tid1-WT. This greater inhibition likely reflects the lack of D-loop formation that could be attributed to Tid1-WT, partially compensating the inhibition of Rad54. Thus, these data together show that Tid1 inhibits D-loop formation in an ATPase-independent and a concentration-dependent manner.

The inhibition of D-loops by Tid1 is independent of the type of substrate used for D-loop formation. D-loops formed with ds98-*607*-78ss substrate having a heterologous 78 nt 3′-flap, led to a 2.5-fold drop in the D-loops, slightly less inhibition than observed with ds98-*607* substrate having a fully homologous 3′-end (*Figure 1C*, *Figure 1—figure supplement 1C*). With the Tid1-KR titration, the largest D-loop inhibition of sixfold was recorded with both substrates (*Figure 1E*, *Figure 1—figure supplement 1D*). The slight differences in the extent of D-loop inhibition among the two substrates were modest and statistically insignificant. Note that the ds98-*607*-78ss substrate led to more efficient D-loop formation (~40% D-loops *versus* ~20% D-loops with ds98-*607*) (*Figure 1—figure supplement 1C,D*) in the absence of Tid1, since D-loops formed with a 3′-flap tend to be more stable (*Wright and Heyer, 2014*). The 40% efficiency in D-loop reactions with supercoiled donor is comparably high, despite the lower stability of linear D-loops. Thus, the inhibition of Rad51-Rad54 mediated D-loop formation by Tid1 is evident irrespective of having a 3′-flap or increased D-loop stability.

Although D-loop reactions with linear dsDNA are known to have low D-loop formation efficiency (*Wright and Heyer, 2014*) compared to supercoiled dsDNA, linear dsDNA was used to prevent any topological constraints imposed by supercoiling. This facet becomes important in the further analysis determining D-loop length, as D-loop length is affected by topological constraints imposed by a negatively supercoiled donor (*Sneeden et al., 2013*). Nevertheless, Tid1-KR inhibited D-loop formation by Rad51 and Rad54 even with supercoiled dsDNA donors, resulting in a 2.5-fold decrease (*Figure 1—figure supplement 1F,G*). This indicates that the inhibition is independent of dsDNA topology or restriction in D-loop length.

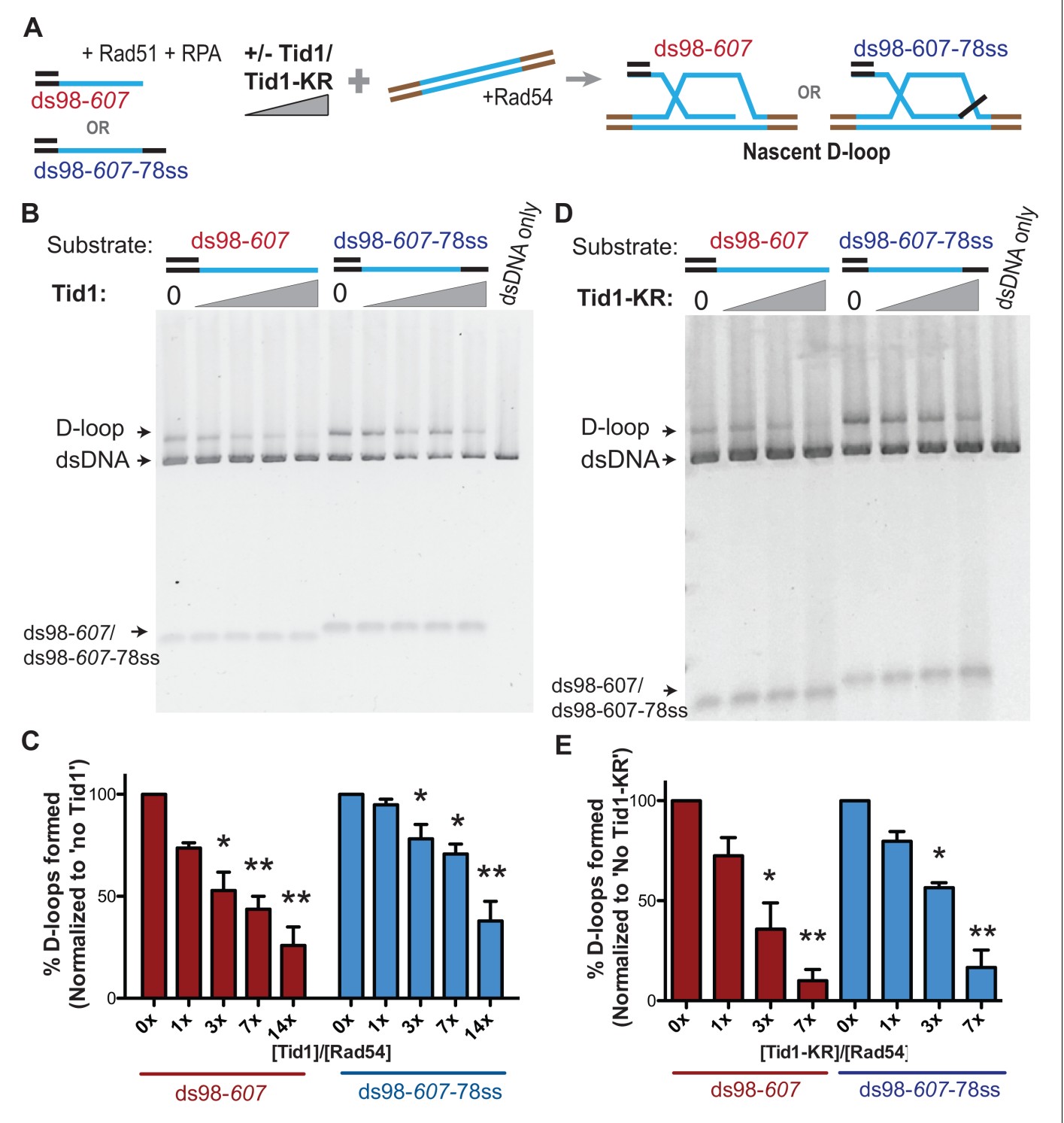

**Figure 1.** Tid1 inhibits D-loop formation in vitro in a concentration-dependent and an ATPase-independent manner. (**A**) Reaction scheme for in vitro D-loop formation assays in presence of Tid1 or the ATPase-defective Tid1-K318R. Here and in all subsequent figures, unless otherwise stated, incubations were in the following order: ssDNA and Rad51 for 10 min, then RPA for 5 min, followed by Tid1/Tid1-KR for 5 min, finally a linearized dsDNA and Rad54 for 15 min (for details, see Materials and methods). Homology between the ssDNA and dsDNA is indicated in blue. (**B, D**) D-loop reactions performed as described in (**A**) with increasing concentrations of Tid1 or Tid1-KR, respectively. The gels were stained with SYBR gold. (**C, E**) Quantitation of the D-loops from the gels in (**B**) and (**D**) as the percentage of donor invaded by ssDNA, respectively. The D-loops are normalized to the

*Figure 1 continued on next page*

Figure 1 continued

amount formed in absence of Tid1/Tid1-KR for each paired reaction. Error bars indicate mean ± SD (n = 3). * indicates p-value<0.05, **<0.005, with a two-tailed t-test, in comparison to 'No Tid1/Tid1-KR' sample. Refer to *Figure 1—figure supplement 1* for absolute D-loop values.

The online version of this article includes the following figure supplement(s) for figure 1:

**Figure supplement 1.** Tid1 by itself stimulates Rad51-mediated D-loops, while inhibits Rad54-mediated D-loops.

These in vitro observations of Tid1 mirror the in vivo observations by *Piazza et al., 2019*, where Tid1 negatively affects the D-loop signal in an ATPase-independent manner. Tid1 is also reported to have ATPase-dependent roles in the repair process, altering the non-crossover frequency and the repair efficiency in cells (*Piazza et al., 2019*). These observations suggest a dual role of Tid1 in somatic HR, with an ATPase-independent effect on D-loops and an ATPase-dependent consequence on the repair outcome. In order to clearly distinguish from its potential ATPase-dependent roles (see Discussion), we continued to employ Tid1-KR for further experiments.

## Tid1 competes with Rad54 to inhibit D-loops

To address whether Tid1 exerts its inhibitory function by directly competing with Rad54, we titrated Rad54 in the D-loop reaction and asked if the amount of Tid1 needed for inhibition titrates with the Rad54 concentration present. We titrated Tid1-KR in the D-loop reaction with either 7.5, 15, or 30 nM Rad54 (*Figure 2A*) and found that Tid1-KR inhibits D-loops in a concentration-dependent manner relative to the amount of Rad54 present (*Figure 2B*). In the D-loop reaction, 30 nM Rad54 requires ~60 nM Tid1-KR to inhibit D-loop formation by 50%, whereas D-loops formed by 7.5 nM Rad54 requires only ~16 nM Tid1-KR to observe 50% reduction (*Figure 2B*). Hence, a 50% reduction in D-loop formation is achieved with a twofold molar excess of Tid1 over Rad54. Tid1 thus competes directly with Rad54.

To confirm this observation and gain insight into the inhibition mechanism, we changed the order of addition of proteins in the reaction, such that Tid1-KR was added prior, simultaneously, or after addition of Rad54 (*Figure 2D and E*, *Figure 2—figure supplement 1A,B*). The inhibition by Tid1-KR was diminished when added at the same time as Rad54 and had no effect when added 10 min after Rad54 and the donor. These results indicate that Tid1-KR inhibits D-loop formation by competing with Rad54 for binding to the Rad51-RPA filament. Once Rad54 is bound to Rad51-RPA filament, Tid1-KR cannot displace it. Moreover, we conclude that Tid1-KR cannot dismantle D-loops after they are formed.

D-loop formation by Rad54 requires its ATPase activity in vivo (*Onaka et al., 2016*) and in vitro (*Tavares et al., 2019*). The Rad54 ATPase activity is stimulated by dsDNA-Rad51 (*Kiianitsa et al., 2002*). To address whether Tid1 interferes with Rad54 ATPase activity, which is essential for inducing Rad51-based DNA strand invasion, we determined the rate of ATP hydrolysis by Rad54 in the presence of dsDNA, Rad51 and increasing amounts of Tid1-KR added either prior to (*Figure 2—figure supplement 1C*) or after Rad54 (*Figure 2—figure supplement 1D*). Tid1-KR inhibited the Rad54 ATPase activity in the presence of dsDNA and Rad51 by sixfold, in a concentration-dependent manner, when added prior to Rad54 (*Figure 2—figure supplement 1C*). However, when Tid1-KR was added after Rad54, the Rad54 ATPase activity was unaffected (*Figure 2—figure supplement 1D*). The ATPase activity arising from Tid1-KR and Rad51 on dsDNA were below the detection limit (*Figure 2—figure supplement 1C*). The Rad54 ATPase activity is stimulated by Rad51 (*Figure 2—figure supplement 1C*), as previously observed (*Kiianitsa et al., 2002*). Note that the concentration of all proteins combined was sub-saturating to the dsDNA, and so that Tid1-KR and Rad54 were not competing for binding to the dsDNA. Thus, the inhibition of Rad54 ATPase activity by Tid1-KR is observed when Tid1-KR is allowed to interact with Rad51 prior to Rad54. In order to eliminate the possibility that Tid1-KR binds dsDNA so strongly as to inhibit Rad54 translocation, we tested the Rad54 ATPase activity on dsDNA in the absence of Rad51 with a Tid1-KR titration (*Figure 2—figure supplement 1E*). At the highest Tid1-KR concentration, the Rad54 ATPase activity is reduced only by ~30% in the absence of Rad51, which is almost insignificant compared to the six-fold inhibition seen in the presence of Rad51. Thus, Tid1-KR inhibits Rad54 translocation specifically in the presence of Rad51. Together, these data suggest that Tid1 competes with Rad54 for binding to the Rad51 filament, thus inhibiting the downstream activation of Rad54 by Rad51 and D-loop formation.

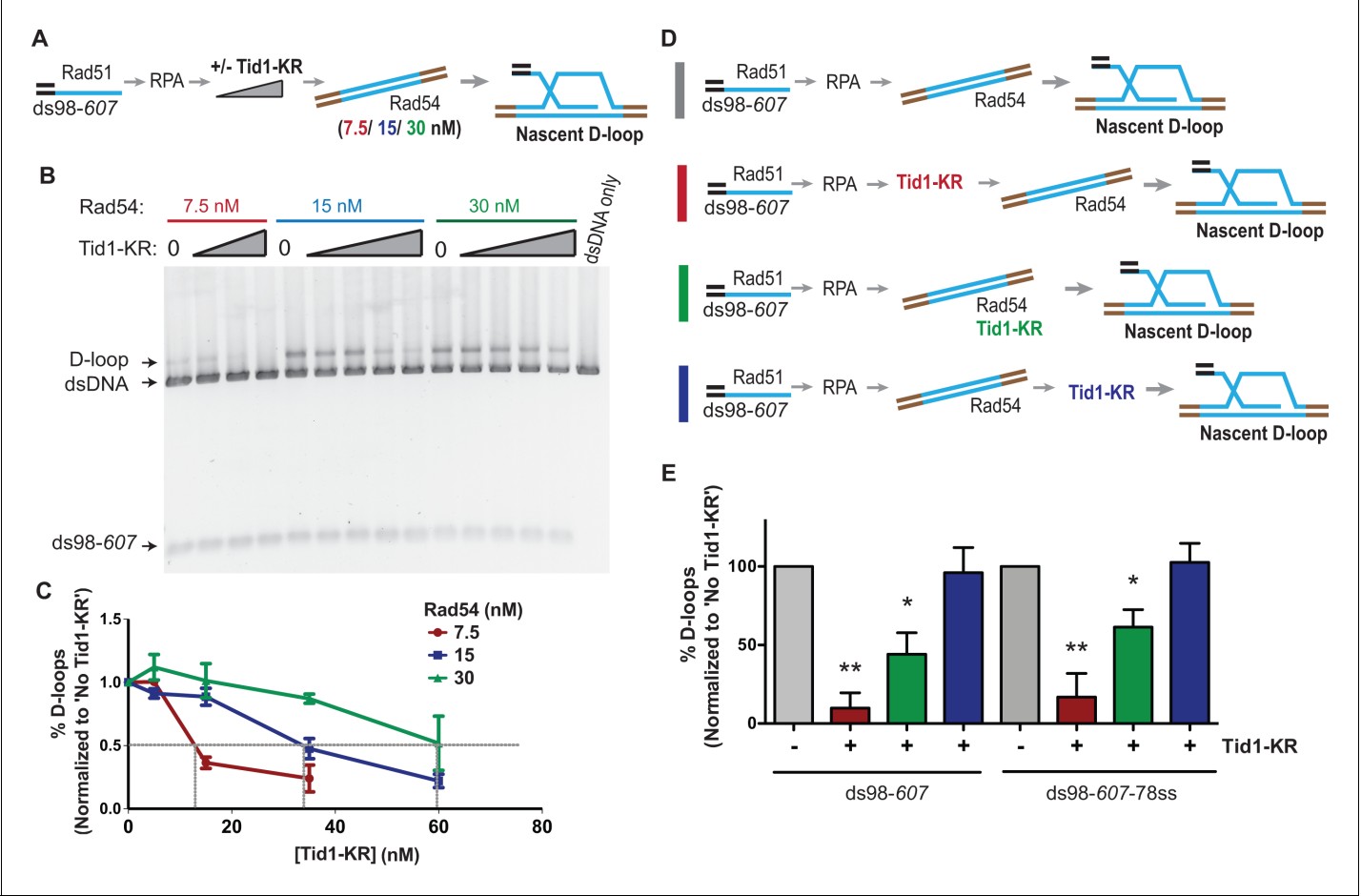

**Figure 2.** Tid1 competes with Rad54 activity and does not inhibit D-loops after they are formed. (**A**) Reaction scheme depicting various Rad54 concentrations, along with Tid1-KR titration in an in vitro D-loop assay. (**B**) SYBR gold stain of gel showing D-loop reaction performed with varying Rad54 and Tid1-KR titrations as indicated in (**A**). (**C**) Quantitation of the D-loops from the gel with normalization to D-loops formed in absence of Tid1-K318R for each of the Rad54 concentrations Error bars indicate mean ± SD (n = 3). Gray lines are drawn to indicate 50% inhibition. (**D**) Different reaction schemes based on the timing of addition of Tid1-KR is indicated by different colored bars on the left. Gray bar indicates D-loop reaction performed in absence of Tid1-KR. Red bar indicates Tid1-KR added to the reaction 5 min before adding dsDNA and Rad54. Green bar indicates Tid1-KR added at the same time as dsDNA and Rad54, whereas blue bar indicates Tid1-KR added 10 min after dsDNA and Rad54. All these reactions were performed using both ds98-*607* or ds98-*607*-78ss substrates. (**E**) Quantitation of D-loops formed as in (**D**) with normalization to the D-loop levels formed in absence of Tid1-KR for each paired reaction. The color of the bars in the graph correspond to the colored bars in (**D**) and represent the respective D-loop samples. Error bars indicate mean ± SD (n = 3). * indicates p-value<0.05, **<0.005, with a two-tailed t-test, in comparison to 'No Tid1/Tid1-KR' sample. Refer to *Figure 2—figure supplement 1B* for unnormalized D-loop values.

The online version of this article includes the following figure supplement(s) for figure 2:

**Figure supplement 1.** Tid1 competes with Rad54 to inhibit D-loops before they are formed and inhibits Rad54's ATPase activity.

## Differential Tid1 abundance between haploid and diploid cells regulates nascent D-loop levels

Several studies have shown that Tid1 is differently expressed in haploid and diploid cells (*Bronstein et al., 2018*; *de Godoy et al., 2008*; *Galitski et al., 1999*). We confirmed this differential expression by comparing the Tid1 protein levels in haploid and diploid cells (*Figure 3A,B*). The steady state Tid1 protein levels are fourfold lower in diploid compared to haploid cells. This suppression in expression was dependent on *MAT*-heterozygosity, as expected from *Nagaraj et al., 2004*. A haploid *MATa* cell transformed with a *MATα*-expressing plasmid showed similar Tid1 levels as a diploid cell. Conversely, a haploid *MATα* cell transformed with a *MATa*-expressing plasmid also had Tid1 protein levels equivalent to in a diploid strain. This supports the observation that the MATa1-α2 repressor binds to the *TID1* promoter and represses its expression in mitotically-dividing diploid

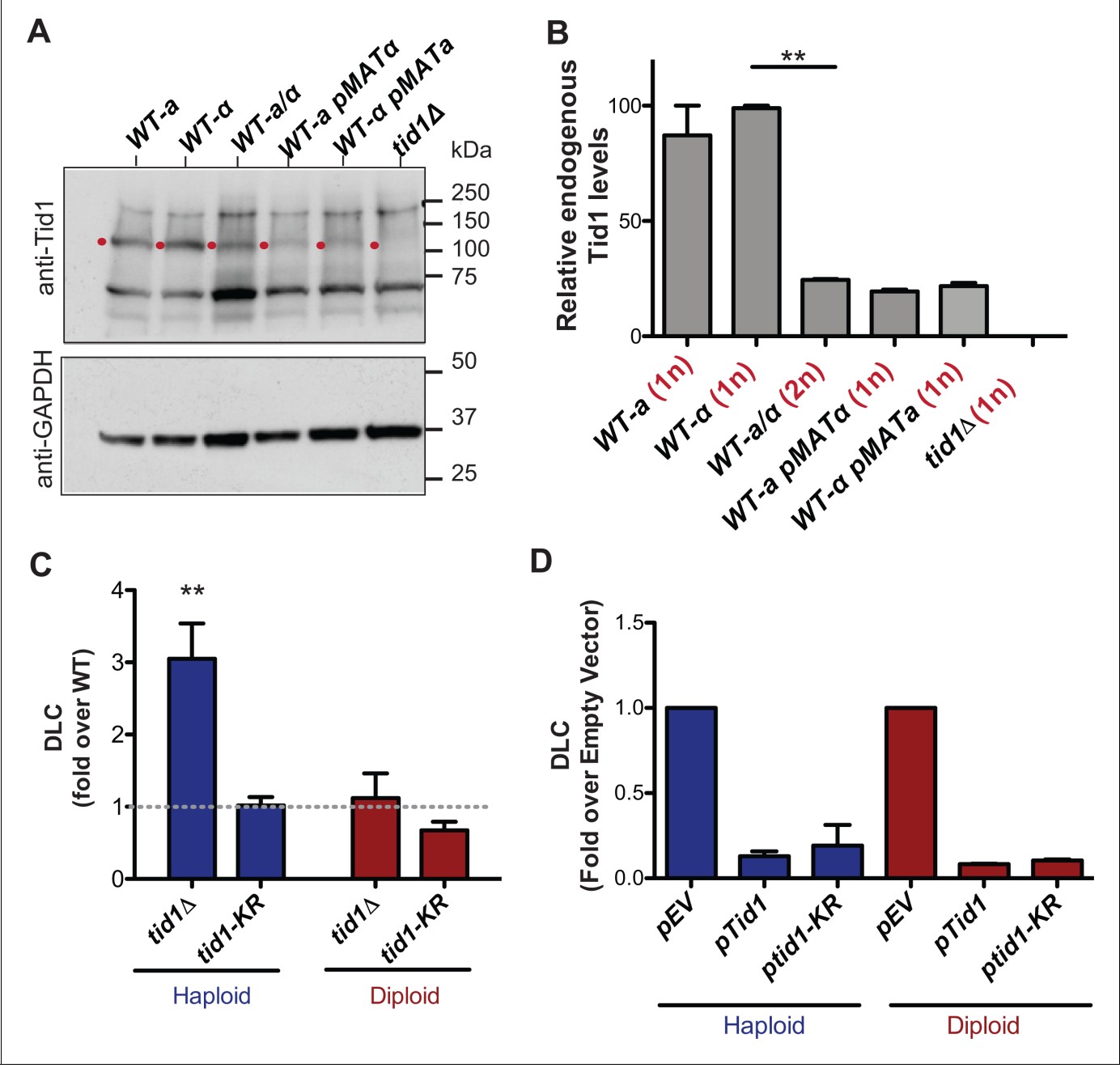

**Figure 3.** Tid1 affects D-loops in vivo in a concentration dependent manner. (**A**) Western blots showing staining with anti-Tid1 or anti-GAPDH antibodies. The band depicting endogenous Tid1 protein (107.9 kDa mol. wt.) is indicated by a red dot. The position of Tid1 was validated by comparison with purified Tid1 (not shown). The other bands are non-specific bands from antibody staining. WT represents wild-type yeast, along with its mating type status. *WT-a pMATα* indicates *MATa* haploid yeast transformed with *MATα*-expressing plasmid. (**B**) Quantitation of endogenous Tid1 levels normalized to the Tid1 levels in haploid *WT-α* strain. It shows *MAT*-heterozygosity dependent suppression of Tid1 expression. Error bars indicate mean ± SD (n = 3). ** p-value<0.005 with two-tailed t-test. (**C**) The D-loop Capture (DLC) signal obtained from the DLC assay is normalized to *WT* DLC signal (indicated by gray dotted line) for both haploid and diploid yeast. The DLC signal was measured 2 hr post DSB induction. Error bars indicate mean ± SD (n = 5). The haploid data is adopted from *Piazza et al., 2019*. (**D**) DLC signal obtained from the DLC assay performed in haploid and diploid yeast transformed with plasmids. *pEV, pTid1* and *pTid1-KR* indicates yeast transformed with an empty vector, a vector containing GST-Tid1 or GST-Tid1-KR under a galactose promoter respectively. The DLC signal was normalized to the sample containing empty vector. The DLC signal was measured 2 hr post DSB induction with galactose. Mean ± SD (n = 2).

cells (**Nagaraj et al., 2004**). Tid1 is thus one of the rare proteins involved in HR to be downregulated in diploid compared to haploid cells. These observations suggest a haploid-specific role for Tid1 in HR.

To address this possibility, we determined nascent D-loop levels in haploid and diploid cells upon site-specific DSB induction using the D-loop Capture (DLC) assay, as described in **Piazza et al., 2019**. Unlike haploids, where *tid1Δ* showed a fourfold DLC increase compared to wild-type cells (**Piazza et al., 2019**), no such increase was observed in diploid cells 2 hr post-DSB-induction (**Figure 3C**). The ATPase-defective mutant *tid1-KR* had no effect on DLC signal in both haploid and diploid cells. Hence, the nascent D-loop inhibition exerted by Tid1 in an ATPase-independent fashion is specific to haploid cells.

To address whether this haploid-specific function of Tid1 solely results from its differential expression level, we transformed haploid and diploid yeast cells with a plasmid containing GST-Tid1 or GST-Tid1-K318R under a galactose-inducible promoter. GST-Tid1 or GST-Tid1-K318R are overexpressed in these cells only when galactose is added to the media to induce DSBs. Two hrs following simultaneous DSB-induction and Tid1 overexpression, a ten-fold drop in the DLC signal was observed both in haploid and diploid cells (**Figure 3D**). The DLC signal was also equally diminished with overexpression of ATPase-dead Tid1-K318R (**Figure 3D**). Hence, the ATPase-independent inhibition of D-loops by Tid1 depends on its abundance in the cell, in line with our in vitro observations.

## A single-molecule assay to define heteroduplex DNA location and length

The DLC assay requires D-loop stabilization by psoralen-mediated crosslinking. Given the estimated inter-strand crosslink density of ~1/500 bp (**Oh et al., 2009**; **Piazza et al., 2019**), this assay cannot unambiguously distinguish between a single, long D-loop and several shorter D-loops comprising the same total heteroduplex length, provided that the total hDNA length remains below the typical crosslink density. Consequently, Tid1 may either alter the absolute number of D-loops in the cell population, as suggested from our in vitro experiments, and/or the average length of hDNA in each D-loop.

To address the possibility of an effect of Tid1 on D-loop length, we developed the DMA to map D-loop length and position at single-molecule level with a near base-pair resolution in vitro (**Shah et al., 2020**). Subsequent analyses of the distribution of D-loop lengths and position in a population of D-loops reveals any biases. DMA employs bisulfite modification of D-loops under non-denaturing condition to deaminate cytosines on the single-stranded regions of the DNA. Thus, cytosine-to-uracil conversions on the displaced strand leaves a footprint of the D-loop that is revealed by sequencing. Here, we leveraged Pac-Bio sequencing on barcoded kilobase-size amplicons to obtain long-range, single-molecule readouts at very high coverage (**Figure 4A**). Each sequencing read would represent either the top strand (containing the displaced strand) or the bottom strand (paired with invading ssDNA) of the dsDNA donor. Footprints are called using a peak threshold, defined here as requiring at least 40% cytosines converted in a stretch of 50 consecutive cytosines (t40w50), to ensure detection of genuine D-loop footprints above the background conversions from sporadic DNA breathing (**Shah et al., 2020**). The D-loops formed were left unpurified from the uninvaded donor DNA, but deproteinized to remove RPA from the displaced strand before subjecting to DMA. The D-loop footprints observed by DMA will reveal individual D-loop length, their position on dsDNA, and distribution of D-loop population. With the t40w50 threshold, the minimum D-loop length detectable is estimated around ~120–200 nt, depending on the density of cytosines. DMA allows mapping D-loops formed in vitro using various single-stranded DNA substrates and a negatively supercoiled or linear double-stranded donor (**Shah et al., 2020**).

We first conducted our analysis using D-loops formed in vitro from ds98-*931*, ds98-*607* and ds98-*197*-78ss substrates and a linear dsDNA donor, devoid of topological constraints. Footprints from DMA were visualized in the form of a footprint map such as the ones depicted in **Figure 4B, C and D**, representing D-loops formed with ds98-*931*, ds98-*607* and ds98-*197*-78ss substrates, respectively. Only the top strand reads containing a footprint are shown here. Of the total top strand reads analyzed, ~20% contained a D-loop footprint for D-loops formed from ds98-*931* and ds98-*607*. Almost none (<0.5%) of the bottom strand reads had a detectable footprint (**Figure 4—figure supplement 1A**). Thus, the occurrence of footprints was highly strand-specific, with D-loop footprints being from 19- to 97-fold more frequent on the top strand compared to the bottom strand across

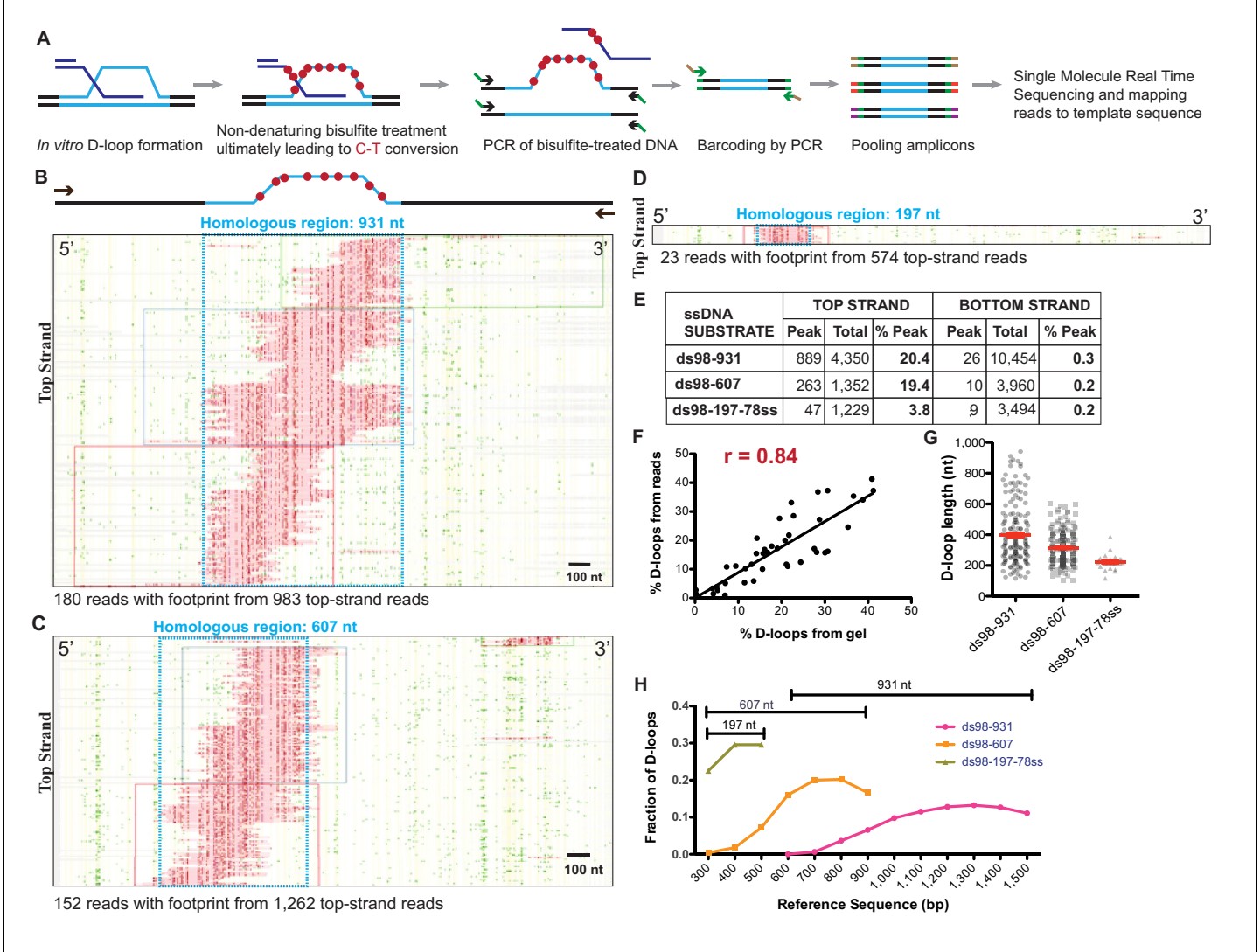

**Figure 4.** Single-molecule D-loop Mapping Assay to map D-loop length, position and distribution in vitro. (**A**) Schematic of the D-loop Mapping Assay using D-loops formed on linear donors in vitro as described in *Figure 1A* as the starting input (for details, see Materials and methods; *Shah et al., 2020*). (**B**) Footprint map depicting reads with a D-loop footprint. Reads are derived from an in vitro Rad51-Rad54-mediated D-loop sample containing ds98-*931* substrate and a linear donor. Only reads from the top strand of dsDNA donor that contain a footprint are shown here. Here and in all subsequent figures with a footprint map, each horizontal line represents one read molecule (or amplicon). The position of each cytosine across the read sequence is indicated by yellow lines. The status of each cytosine along the sequence is color-coded with green representing C-T conversions. The status of cytosine is changed to red if the C-T conversions cross the peak threshold and are thus defined as D-loop footprints. Unless otherwise mentioned the peak threshold is t40w50 (40% cytosines converted to thymine in a stretch of 50 consecutive cytosines). The reads are clustered based on the position of footprints in 5′ to 3′ direction. The faintly colored boxes indicate the clusters. The blue box represents dsDNA region homologous to *931* nt on the invading substrate. Scale bar is 100 nt. For bottom strand reads, refer to *Figure 4—figure supplement 1A*. (**C**) Footprint map depicting reads containing D-loop footprints from an in vitro Rad51- and Rad54-mediated D-loop reaction performed with ds98-*607* substrate and a linear donor. (**D**) Footprint map depicting reads containing D-loop footprints from an in vitro Rad51- and Rad54-mediated D-loop reaction performed with ds98-*197-*78ss substrate and a linear donor. Note that a smaller fraction of reads with footprints was observed for the ds98-*197*-78ss substrate due to its lower D-loop formation efficiency (*Wright and Heyer, 2014*). (**E**) Table summarizing the total number of reads containing a footprint as 'peak' and the total number of reads analyzed as 'total' for each strand. '% D-loops/% Peak' indicate the percentage of reads containing a footprint. The data represents a cumulation from >3 independent replicates. The percentage of D-loops are calculated by dividing the number of reads with footprint by the total number of reads for that strand. Shown are cumulative data from three to five independent replicates, of which one to two overlap with the data reported in the accompanying manuscript (*Shah et al., 2020*). (**F**) Dot plot indicating correlation between the percentage of reads containing D-loop footprint by DMA and the percentage of D-loops seen on the gel relative to the uninvaded dsDNA. Pearson coefficient's r = 0.84 for 42 XY pairs. p-value<0.0001. (**G**) Dot plot indicating individual D-loop lengths measured across substrates with varying homology lengths. D-loop length was measured in nt based on the footprint size called by the DMA assay. The data represents a cumulation from >3 independent replicates. Red error bars indicate mean ± SEM. (**H**) Distribution of D-loop footprints across the region of homology (as indicated by a capped line for each substrate type), with

*Figure 4 continued on next page*

*Figure 4 continued*

an enrichment at the 3′-end. The distribution is measured by binning each footprint in 100 nt bins across the homology non-exclusively and depicted as the percentage of total D-loops within each bin. 'Reference sequence' indicates the position on dsDNA donor.

The online version of this article includes the following figure supplement(s) for figure 4:

**Figure supplement 1.** D-loops formed with various length substrates.

all the three substrates (*Figure 4E*). Moreover, as expected, the footprints fell exactly within the region of homology (boxed in blue) in all three cases. In summary, the data indicate that the D-loop footprints measured using DMA were strand-specific, homology region-specific and substrate-specific.

Importantly, the percentage of top strand reads containing a D-loop footprint (% D-loops from the reads) correlated well with the percentage of D-loops observed on gel assays relative to the uninvaded donor (*Figure 4—figure supplement 1B,C*). In fact, there was a strong 84% correlation between the quantitation of D-loops measured by gel assay and by DMA (*Figure 4F*), when D-loop values from all samples were combined (including samples with Tid1 that are discussed later). Thus, while short, unstable D-loops might be lost either due to threshold detection limits or instability during treatment, the high correlation between the two orthogonal methods suggests that relative D-loop quantification across paired samples is feasible and accurate by DMA.

Interestingly, the distribution of D-loop footprints was not uniform. D-loops varied in length and in their relative positions across the homology (*Figure 4B–D and G*). The two longest substrates, ds98-*931* and ds98-*607*, showed the largest diversity of D-loop lengths spreading across the homology length (*Figure 4G*). In case of the ds98-*197*-78ss substrate, all D-loops were ~200 nt long, restricted by minimum detection length at the lower end and maximum homology length at the upper end. The longest observed D-loop footprints in ds98-*931* and ds98-*607* spanned the entire length of homology, nearing 900 and 600 nucleotides, respectively. These long D-loops spanning the homology comprised a small, yet significant proportion of the total D-loops with the ds98-*931* (5%) and ds98-*607* (7%) substrates. As a consequence of the varied D-loop lengths observed, the average D-loop lengths were proportional with the size of the homology, reaching 410 and 315 nt for the ds98-*931* and ds98-*607* substrates, respectively (*Figure 4G*). Thus, D-loop lengths of various sizes were observed limited only by the detection limit and the homology length. Finally, the D-loop position varied across the region of homology, but the signal was strongly enriched at the 3′-end of the invading DNA (*Figure 4H*), as previously observed (*Wright and Heyer, 2014*). In conclusion, the DMA allows D-loop position and length to be defined with good efficiency, sensitivity and resolution, provided the D-loop is longer than the 120–200 nt limit.

## Tid1 regulates D-loop length

To address whether Tid1 affects D-loop length in addition to D-loop levels, we performed DMA on D-loops formed in the presence of Tid1 or Tid1-KR. We envisioned that Tid1 may alter D-loop lengths by competing with Rad54 in binding to the Rad51 filament interstitially. Rad51 is not highly processive in forming filaments, unlike its bacterial homolog RecA (*Galletto et al., 2006*; *Sanchez et al., 2013*). Hence, Rad51 filaments often retain gaps of Rad51-free ssDNA, that are potential binding sites for Rad54 (*Kiianitsa et al., 2006*; *Sanchez et al., 2013*). Tid1 may also potentially bind at these interstitial sites in the filament via its Rad51 interaction domain (*Petukhova et al., 2000*), and in turn may block the translocation of Rad54 by acting as a physical roadblock, leading to formation of shorter D-loops.

To mimic Rad51 filaments comprising of intermittent gaps, we lowered the Rad51 concentration by fourfold from saturating levels (Rad51: nt = 1:3) to Rad51: nt = 1:12. First, we analyzed if lowering the Rad51 concentration altered D-loop characteristics. D-loop reactions were performed using a linear dsDNA donor, devoid of any topological constraints and a substrate with long homology (~900 nt) to allow modulation of D-loop lengths from the action of recombinant proteins. *Figure 5A and B* show that the D-loop levels were stable for Rad51: nt ratios varying from the usual 1:3 to 1:12. Over-saturation (Rad51: nt = 1:1) or more extreme sub-saturation (Rad51: nt = 1:24) led to a reduction in D-loop levels, similar to prior observations with hRAD51 (*Rossi and Mazin, 2008*). Thus, reducing Rad51 concentration by fourfold (Rad51: nt = 1:12) still maintained efficient D-loop formation. In

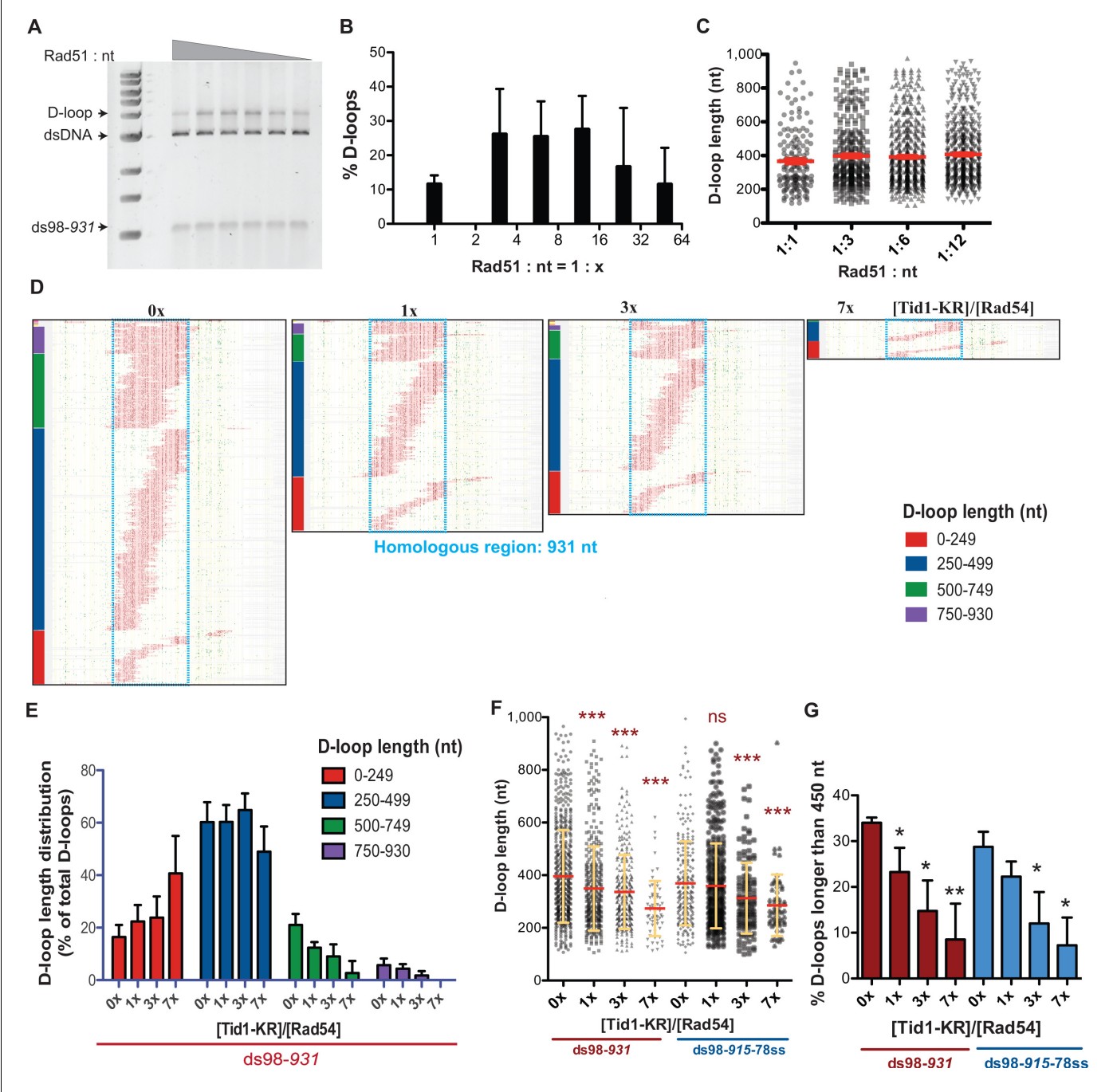

**Figure 5.** Tid1-KR restricts formation of longer D-loops. (**A**) Gel stained with SYBR gold depicting D-loops formed with decreasing concentrations of Rad51. (**B**) Quantitation of D-loops from the gel in (**A**), where 1 Rad51 to 3 nt (Rad51: nt = 1:3) is a saturating Rad51 concentration. Mean ± SD (n = 3). (**C**) Dot plot showing the D-loop lengths of D-loop footprints seen with varying Rad51 concentrations in DMA assay. In red is Mean ± SEM (n = 3). Refer to **Figure 5—figure supplement 1** for the corresponding footprint maps. (**D**) Footprint map of reads containing D-loop footprints in DMA assay. The D-loops were formed from a ds98-*931* substrate and Rad51: nt = 1:12 concentration, followed with Tid1-KR titration. The reads were clustered based on D-loop length and position. The colored bars on the left indicate four different length clusters (0–249, 250–499, 500–749 and 750–930 nt). The footprints for each sample depict a cumulation from four independent replicates. (**E**) Quantitation of the distribution of D-loop lengths within each length cluster as a percentage of total D-loops for each sample in (**D**). The color of the bars correlates with the cluster bars on the footprint maps in (**D**). Mean ± SD (n = 4). (**F**) Dot plot depicting the D-loop lengths observed by DMA, from both ds98-*931* and ds98-*915*-78ss substrates. In red is mean ± SEM, in yellow is mean ± SD (n = 4). *** indicates p-value<0.0005, with a two-tailed paired t-test, in comparison to 'No Tid1-KR' sample. (**G**) Percentage of D-loop footprints that are longer than 450 nt from samples in (**F**). Mean ± SD (n = 4). * indicates p-value<0.05, **<0.005, with a two-tailed paired t-test, in

*Figure 5 continued on next page*

*Figure 5 continued*

comparison to 'No Tid1-KR' sample. Refer to *Figure 5—figure supplement 2* for the distribution of D-loops formed from ds98-*931 and ds98-915*-78ss substrates. Refer to *Figure 5—figure supplement 3* for D-loop footprints observed in presence of WT Tid1.

The online version of this article includes the following figure supplement(s) for figure 5:

**Figure supplement 1.** Changes in Rad51 concentration does not significantly alter D-loop characteristics in the D-loop Mapping Assay.

**Figure supplement 2.** Tid1-KR limits D-loop length but does not alter the distribution of D-loop position.

**Figure supplement 3.** Tid1 limits D-loop length.

parallel to gel assays, we measured D-loop formation by DMA under the same conditions (*Figure 5—figure supplement 1A*), which revealed that D-loop efficiencies detected by DMA (*Figure 5—figure supplement 1B,C*) correlated well with the gel-based measurements (*Figure 5B*). Moreover, the distribution of D-loop lengths or the average D-loop length did not change significantly between the 1:3 and 1:12 Rad51: nt stoichiometric ratios (*Figure 5C*). Similarly, there was no significant difference in the distribution of D-loop position across the region of homology (*Figure 5—figure supplement 1D*). Thus, the Rad51: nt stoichiometric ratio can be lowered from 1:3 to 1:12 without significantly altering D-loop levels, lengths, position, or distribution, while potentially providing intermittent binding sites for Tid1 or Rad54 in the pre-synaptic or post-synaptic filament.

We next tested the effect of Tid1-KR titration on D-loops formed from such undersaturated Rad51 filaments. Two substrates with long homologies, ds98-*931* and ds98-*915*-78ss, were used to provide a long window for D-loop length alterations. Again, Tid1-KR inhibited D-loop formation under these Rad51 conditions (*Figure 5—figure supplement 2A,B*), as previously seen with Rad51: nt ratio of 1:3. *Figure 5D* and *Figure 5—figure supplement 2C* depict the footprint maps of D-loop footprints observed by DMA under increasing concentrations of Tid1-KR for the ds98-*931* and ds98-*915*-78ss substrates, respectively. For each footprint map shown, footprints from >3 independent replicates were pooled to remove any sampling bias in the analysis. In agreement with measurements derived from the gel-based assays (*Figure 5—figure supplement 2E*), the DMA assay shows a ten-fold inhibition of D-loops by Tid1-KR (*Figure 5—figure supplement 2C,D*). The inhibitory effect of Tid1-KR on D-loop levels was independent of the presence of a 3'-heterology on the invading substrate (*Figure 5—figure supplement 2C–E*), similar to previous observations. Thus, these observations confirm the concentration-dependent and ATPase-independent inhibition of D-loop formation by Tid1.

To visualize the effect of Tid1-KR on D-loop size, D-loop footprints were clustered based on their size, along with their position (*Figure 5D*, *Figure 5—figure supplement 2C*). The D-loop sizes were clustered into four categories: <250 nt, 250–499 nt, 500–749 nt, and >750 nt. The percentage of D-loops falling into each length category was quantified and depicted in *Figure 5E* for the ds98-*931* and in *Figure 5—figure supplement 2F* for the ds98-*915*-78ss substrate. Note that the percentage of D-loops in each length category was normalized to the total D-loops detected for that sample, to allow direct comparison of length distributions. Increasing Tid1-KR concentration led to a decrease in the proportion of longer D-loops with a concomitant increase in the proportion of shorter D-loops. Consequently, the average D-loop lengths also decreased by 1.5-fold (*Figure 2F*) from 410 nt to 270 nt. A sharp and significant four-fold decrease was observed in D-loops longer than the average length of untreated D-loops that is larger than 450 nt (*Figure 5G*). We note that the observations on D-loop length were robust even when peak calling parameters were lowered to permit the detection of smaller D-loops (*Figure 5—figure supplement 2G,H*). At the lower t25w20 threshold, more D-loops of shorter lengths were observed, as expected. Yet, longer D-loops were progressively lost with increasing Tid1-KR concentration. Lastly, the alterations in D-loop lengths did not affect the overall distribution of D-loop position across the region of homology (*Figure 5—figure supplement 2I,J*). Thus, Tid1-KR significantly reduced the length of D-loops formed along with a decrease in D-loop levels, while the position of D-loops remained unaffected.

Similarly, D-loops treated with increasing concentration of Tid1-WT led to an enrichment of shorter D-loops and loss of longer D-loops, as evident from the D-loop footprint maps (*Figure 5—figure supplement 3A*) and quantification of D-loop length distributions (*Figure 5—figure supplement 3B*). As expected, increasing Tid1 concentrations caused a decrease in average D-loop lengths (*Figure 5—figure supplement 3C*) and a reduction of the D-loop footprint levels (*Figure 5—figure*

*supplement 3D*). In all cases, D-loops were specific to the top-strand of the donor DNA (*Figure 5— figure supplement 3E*). Thus, both Tid1 and Tid1-KR promote formation of shorter D-loops along with a reduction in D-loop levels, making it unlikely that the outcomes are due to a poisoning effect of Tid1-KR being stuck on DNA. Together, these results support the hypothesis that Tid1 not only competes with Rad54 and inhibits D-loop formation, but Tid1 may also block Rad54 translocation resulting in shorter D-loops.

### In vivo mating type switching regulation by Tid1

Since Tid1 is specifically expressed in haploid yeast, with an effect on D-loops seen only in haploids, we wondered if Tid1 plays a role in mating type switching that is specific to haploid cells. D-loop regulation by Tid1 may be important during mating type switch in two potential ways. First, in *MAT-alpha* cells, Tid1 may promote invasion into *HMRa* (with 239 nt homology at *Z*-end and 703 nt at *WX*-end) that has shorter homologies than *HMLalpha* (with 327 nt homology at *Z*-end and 2,180 nt at *WX*-end). Shorter D-loops promoted by Tid1 may increase the likelihood of invasion into the donor with opposing mating type. This might be irrelevant in the case of *MATa* cells, where the recombination enhancer (RE) may dominate the invasion into *HMLalpha.* Second, the Z-end is proposed to be the dominant invading end (*Hicks et al., 2011*). Since the Z-end has much shorter homology than the *WX*-end, the effect of Tid1 on D-loop length may be further influencing the choice of using the Z-end for invasion. The explanation that a non-homologous flap at the *WX*-end is minimizing invasion from that end is insufficient in the case of a fully homologous donor. Hence, Tid1 via its effect on D-loops may promote invasion from the Z-end. To test the possibility that Tid1 influences donor choice based on the length of homology, we used strains (*Mehta et al., 2017*) designed to have either a 148 nt or 2,216 nt homology at *HML* donor to the Z-end of *MAT* (*Figure 6A*) and created *TID1* deletion mutants. In these strains, the fully homologous *HMR* locus was deleted. Invasion was studied by detecting the initiation of DNA synthesis on *HML* by a primer extension assay (*Mehta et al., 2017*).

We found that with 2216 bp long homology, *tid1Δ* mutants exhibited accelerated kinetics of D-loop extension at the *HML* donor (*Figure 6B*). Conversely, with short homology of 148 bp, *tid1Δ* mutants showed a slight but significant decrease in the kinetics of D-loop extension (*Figure 6C*). These results suggest that presence of Tid1 in haploids both inhibits the usage of long homologies and promotes usage of short homologies for HR repair, at least in the context of *MAT*. Hence, Tid1 may participate in inhibiting the use of the fully homologous donor to promote mating-type switching.

To further test the importance of D-loop length restriction by Tid1 during mating-type switching, we analyzed cell viability. Again, strains with either 148 or 2216 bp homology to the Z-end of *MAT* were used to determine viability after DSB induction at *MAT*. Deletion of *TID1* led to a 20% decrease in cell viability post-DSB induction in strains having long 2216 bp homology at the Z-end (*Figure 6D*). However, shorter homologies of 148 bp resulted in no change in viability in absence of Tid1 (*Figure 6D*). The strain with 148 bp homology had ~80% viability under wild-type conditions, as previously seen by *Mehta et al., 2017*, where reduction in homology length reduces viability after DSB induction. However, *tid1Δ* mutants did not further reduce the viability. Thus, Tid1 is required to maintain cell viability post double-strand break formation for recombination events involving long, 2000 bp homologies, in the context of mating type switching. *Piazza et al., 2019* also similarly observed a decrease in viability in *tid1Δ* mutants using an ectopic recombination system. Here, we show the effect of Tid1 directly in the context of mating-type switching.

## Discussion

### A novel D-loop Mapping Assay (DMA) to map D-loop length, position, and distribution

We developed a novel DMA assay (*Shah et al., 2020*) to map individual D-loop characteristics such as D-loop length and position at near base-pair resolution, and their distribution among a population of D-loops. DMA also allows relative comparison of D-loop levels that correlate well with the gel-based detection method. The assay is robust, sensitive, and provides high resolution on D-loops. Here, we used this assay and showed that it responds to changes in D-loop characteristics by D-loop

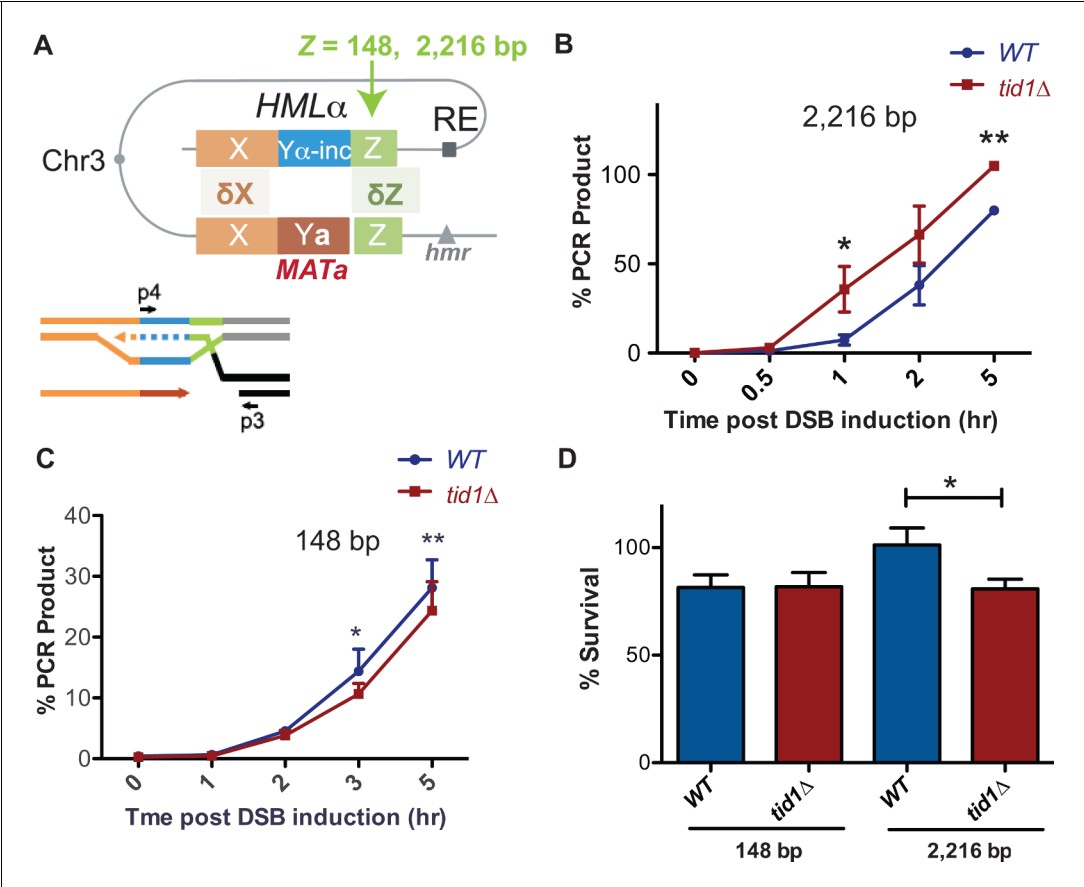

**Figure 6.** Tid1 affects kinetics of D-loop extension and cell survival depending on the length of homology between the donor and *MAT* during mating type switch. (**A**) Schematic of the primer extension assay adopted from *Mehta et al., 2017*. Homology length to the invading Z-end (green) was altered at *HML* to be either 148 bp or 2216 bp. Arrowhead indicates *HO-cut site*. Primers (indicated by black arrows as p3 and p4) specific to the newly synthesized DNA (shown by dotted lines) after strand invasion into *HML* was used to quantify extended D-loops by the primer extension assay. (**B, C**) Kinetics of new DNA synthesis post D-loop formation as measured the primer extension assay. Graphs show qRT-PCR product at intervals post HO endonuclease induction by galactose in strains with 148 bp or 2216 bp homology at the Z-end. The amount of PCR product obtained from a switched *MATα-inc* colony was set to 100%. The Cp values were normalized to Arginine PCR product formation as in *Mehta et al., 2017*. Mean ± SD (n = 3). (**D**) Viability of strains having 148 bp or 2216 bp homology at the Z-end between *HML* and *MAT* loci after HO endonuclease induction by galactose. Mean ± SD (n ≥ 3). * indicates p-value<0.05 with paired two-tailed t-test.

modulators. The assay is widely applicable to study in vitro D-loops formed with a variety of different substrates and dsDNA topologies. The assay may allow understanding an interplay of various proteins factors affecting D-loop formation and disruption that maybe derived from various species. (*Shah et al., 2020*).

## The mechanism of Tid1 in Rad51- and Rad54-mediated D-loop formation

HR is a dynamic and complex pathway with multiple protein players and regulators ensuring repair fidelity. Regulation of D-loops may play a vital role in influencing donor choice, as well the repair outcome and fidelity (*Piazza and Heyer, 2019*). The effect of Tid1 on D-loops was recently described in vivo in somatic HR repair (*Piazza et al., 2019*). Absence of Tid1 lead to a four-fold increase in the D-loop signal measured by the DLC assay 2 hr post break-induction. This negative effect of Tid1 on D-loops was independent of its ATPase activity, while downstream steps such as D-loop extension and promotion of the non-crossover outcome of HR were ATPase-dependent (*Piazza et al., 2019*). D-loops formed in somatic cells differ from their meiotic counterparts in that they lack the meiosis-specific recombinase Dmc1, which preferentially interacts with Tid1 rather than Rad54 (Nimonkar et al. 20112). By contrast, somatic D-loops are predominantly formed by Rad51-

and Rad54-mediated activity. Thus, the role of Tid1 in Rad51- and Rad54-mediated recombination was unclear.

Based on the experiments presented here, we reach the following conclusions regarding the mechanism of action of Tid1 in HR at the nascent D-loop level:

1. Tid1 inhibits Rad51- and Rad54-mediated D-loop formation in vitro in an ATPase-independent manner by competing with Rad54 for binding to the Rad51 filament (*Figures 1* and *2*). This effect is independent of the topology of the donor (topology-free linear dsDNA and supercoiled dsDNA), as well as the structure of a D-loop (with a free 3′-end or a non-homologous 3′-flap). Hence, it is unlikely, that Tid1 diminishes D-loop formation due to its ability to alter DNA topology (*Petukhova et al., 2000*). These data suggest that D-loop stability, structure, and topology do not affect the inhibition by Tid1 on D-loop formation.
2. In addition, Tid1 limits D-loop length, but not the location of hDNA in a homologous region, in an ATPase-independent fashion (*Figure 5*). At higher concentrations, Tid1-KR leads to a drop in average D-loop length from 410 nt to 270 nt for the D-loops formed using ds98-*931* ssDNA. The frequency of D-loops longer than 450 nt is reduced from 40% to 10%. This limitation in length might be a consequence of Tid1 acting as a physical roadblock to Rad54 translocation via the ability of Tid1 to compete with Rad54 in Rad51 binding.
3. Tid1-KR also inhibits Rad54 ATPase activity specifically in the presence of Rad51 (*Figure 4—figure supplement 1*). Thus, it is likely that Tid1, like Rad54, binds to Rad51 filament ends, subsequently blocking Rad54 translocation.
4. Tid1 protein levels are under direct control by the diploid-specific MAT a1/α2 transcriptional repressor (*Figure 3A and B*; *Nagaraj et al., 2004*). This regulation of Tid1 abundance between life cycle phases makes it a haploid-specific negative regulator of D-loops in vivo (*Figure 3C*). In absence of Tid1, a fourfold increase in D-loop signal is seen in haploid yeast (*Piazza et al., 2019*) but not in diploids. However, overexpression of Tid1 leads to a ten-fold drop in the D-loop signal (*Figure 3D*), confirming the concentration-dependent and ATPase-independent effect of Tid1 on D-loops.
5. Lastly, Tid1 promotes D-loop extension at the donor with shorter homology while decreases the kinetics of extension at the donor with long homology (*Figure 6*). This distinction promoted by Tid1 may aid mating type switch in haploid cells. Absence of Tid1 also reduces cell viability post HR-mediated DSB repair involving a long, 2000 bp homology, but the viability is unaltered with short homology (*Figure 6*). This difference in viability may be explained by an accumulation of long, toxic D-loop intermediates that are not easily disrupted. Thus, Tid1 may alter D-loop extension kinetics and cell viability by regulating the D-loop length.

## Model: Tid1 as a roadblock to Rad54

Taking all results into account, we propose a 'roadblock model' for the role of Tid1 in modulating D-loops in haploid yeast (*Figure 7*). We propose that Tid1 acts twofold, to limit D-loop formation and D-loop length. Both result as a consequence of a competition between Tid1 and Rad54 in binding to Rad51 filaments at either the pre-synaptic (Rad51-ssDNA) or post-synaptic state (*Figure 7A*). Rad51 is not as cooperative in bindng DNA as its bacterial homolog RecA (*Galletto et al., 2006*; *Sanchez et al., 2013*) and is prone to leaving Rad51-free gaps in the filament. These gaps are potential binding sites for Rad54 at the pre-synaptic (*Kiianitsa et al., 2006*; *Sanchez et al., 2013*), as well as the post-synaptic state (*Sanchez et al., 2013*; *Tavares et al., 2019*). It is thus plausible that the Rad54 paralog, Tid1, may also bind pre- and/or post-synaptic Rad51 filaments, via its N-terminal Rad51 interaction domain (*Chi et al., 2006*; *Petukhova et al., 2000*; *Santa Maria et al., 2013*), similar to the Rad54 N-terminus (*Raschle et al., 2004*). In line with this, Tid1 is recruited to DSBs within 1 hr of break-induction in a Rad51-dependent manner (*Kwon et al., 2008*). Tid1 is also phosphorylated in response to DNA damage in somatic cells (*Ferrari et al., 2013*), but it remains unclear how phosphorylation of Tid1 alters the Tid1 interaction with Rad51 or its translocation activity. Phosphorylation of Rad54, for instance, suppresses interaction between Rad54 and Rad51 during meiotic recombination (*Niu et al., 2009*). Irrespective of whether Tid1 and Rad54 arrive at the pre-synaptic and/or post-synaptic filament in vivo, relative concentrations of both and their interaction with Rad51 would drive the outcome of the competition with subsequent alterations in D-loop level (*Figure 7B*) and D-loop length (*Figure 7C*).

First, inhibition of D-loop levels may be seen when Tid1 outcompetes Rad54 in binding to the Rad51 filament ends (*Figure 7B*). In the absence of, or at relatively lower Tid1 concentrations (such

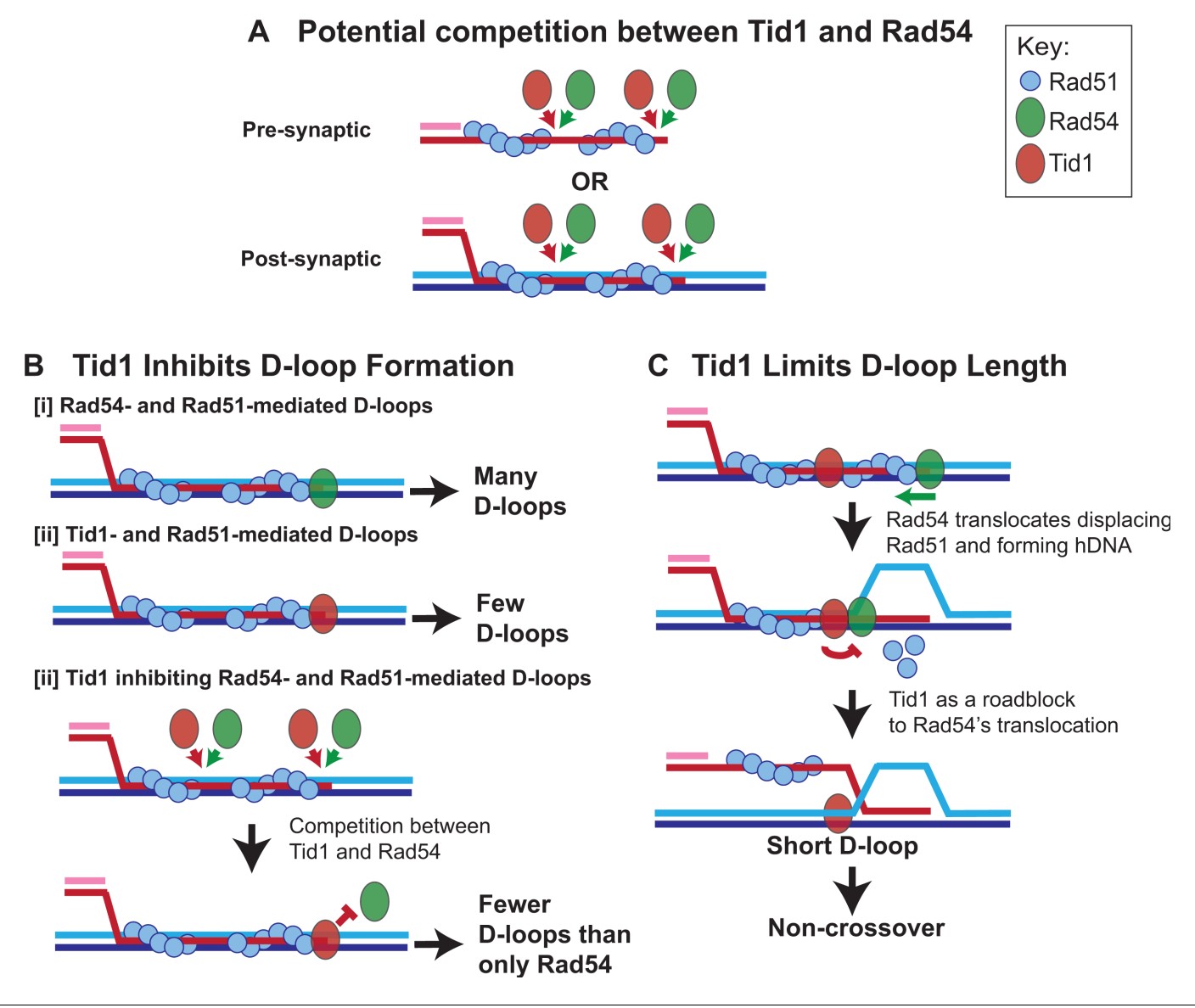

**Figure 7.** Model: Tid1 competes with Rad54 in binding to the filament ends, limits D-loop length and may promote non-crossover outcome. (**A**) Model depicts Tid1 competing with Rad54 for binding to Rad51 filament ends either at pre-synaptic or post-synaptic stage. Since Rad51 is not as cooperative as its bacterial homolog, RecA (*Galletto et al., 2006*; *Sanchez et al., 2013*), gaps in the filament can be expected, that provide potential binding sites for Rad54 and/or Tid1. Both Tid1 (*Chi et al., 2006*; *Petukhova et al., 2000*; *Santa Maria et al., 2013*) and Rad54 (*Raschle et al., 2004*) interact with Rad51 *via* their N-terminal domain and are recruited to DSB sites (*Kwon et al., 2008*). (**B**) Model depicting in vitro D-loop levels when formed in presence of Rad54 and/or Tid1 and Rad51. (**C**) A 'roadblock model' explaining the effect of Tid1 on D-loop length. Tid1 potentially bound intermittently within a pre- or post-synaptic Rad51 filament, can act as a physical roadblock to Rad54 translocation. Rad54 translocation is stimulated by Rad51, simultaneously displacing Rad51 and forming a hDNA (*Wright and Heyer, 2014*). When Rad54 encounters Tid1, the N-terminal domain disengages resulting in the formation of shorter D-loops. Short, dynamic D-loops can subsequently prevent dHJ formation and the possibility of a crossover outcome.

as in diploid cells), Rad54 efficiently forms D-loops (*Figure 7B–I*; *Wright and Heyer, 2014*), as evident by the higher recombination efficiency in diploid cells (*Morgan et al., 2002*; *Mozlin et al., 2008*; *Valencia-Burton et al., 2006*). Tid1 alone stimulates Rad51-mediated D-loop formation, although much less efficiently than Rad54 (*Figure 7B–II*; *Nimonkar et al., 2012*). Hence, with co-presence of Tid1 and Rad54, fewer Rad54-mediated D-loops are formed as Tid1 outcompetes Rad54 from Rad51 filament ends and prevents Rad54 stimulation (*Figure 7B–III*).

Second, restriction of D-loop length may occur if Tid1 also competes with Rad54 (*Sanchez et al., 2013*) to be localized interstitially in the Rad51 filament (*Figure 7C*). The Tid1 ATPase activity is not activated when bound to ssDNA (*Petukhova et al., 2000*). Hence, Tid1 present between the filaments may act as a physical roadblock to Rad54 translocation, resulting in formation of shorter D-loops (*Figure 7C*). Interaction of Rad54 with Tid1 instead of Rad51 through its N-terminal domain may cause Rad54 to disengage (*Wright and Heyer, 2014*), preventing further Rad54-mediated hDNA formation. The model supports the observations found in vitro and in vivo, where Tid1 has an effect on D-loops in a concentration-dependent and translocation-independent manner. Based on the model, long homology allows a bigger window for Tid1 incorporation between Rad51 filaments. Such an inhibitory activity of Tid1 on D-loops depending on the homology length might be especially beneficial to haploid cells as discussed below. Thus, the model provides a mechanistic explanation for the antagonistic relationship between the two paralogs. This view is consistent with recent single-molecule data that led to the conclusion that Rad54 and Tid1 exert independent and distinct functions (*Crickard et al., 2020*).

## Physiological relevance of D-loop modulation by Tid1

The physiological relevance of the effect of Tid1 on D-loops in haploid yeast is two-fold. First, mating type switch in *MATalpha* haploid yeast requires D-loop formation at the *HMRa* donor with shorter homology (239 or 703 nt), than the *HMLalpha* donor with longer homologies (327 or 2,180 nt). Moreover, irrespective of the mating-type, the invasion needs to be regulated such that the preferred *Z*-end with short (~300 nt) homology forms D-loops rather than the *WX*-end with long homology (~1,400 nt), to allow invasion into the donor with an opposing mating type (*Haber, 2012*). In other words, formation of long, stable D-loops on fully homologous donor would counteract the mating type switch, requiring a mechanism to restrict D-loop size and to allow better chance of invading shorter homology donor. We show that Tid1 promotes invasion and subsequent D-loop extension in the donor with short homology (148 bp), while making extension at donor with long homology (2216 bp) relatively less efficient (*Figure 6A–C*). Another mechanism by which Tid1 may promote mating-type switch is that the D-loop characteristics may determine whether the broken *MAT* molecule invades a potentially unbroken fully homologous sister chromatid or the intrachromosomal *HML/HMR* loci. Natural levels of HO-endonuclease may raise the possibility that both sister chromatids may not be cleaved at the same time. However, *Klein, 1997* showed that Tid1 inhibits intra- and inter-chromosomal recombination by ~two-fold. Hence, it seems unlikely that Tid1 promotes intrachromosomal recombination over sister chromatid recombination. Therefore, we propose that by restricting the D-loop length, Tid1 may prevent formation of long, stable D-loops in the fully homologous donor and thus, may aid mating type switching.

Second, we have shown that deletion of Tid1 leads to a loss of non-crossover outcomes in haploid yeast (*Piazza et al., 2019*), as well as a decrease in the total repair efficiency. The decrease in repair efficiency observed in *tid1Δ* is dependent on the length of homology. In assays involving short homology (0.5 kbp) with the donor, there is no effect of *tid1Δ* mutants on the repair efficiency compared to wild-type cells. While, with 5.6 kb long homology donors, there is a 40% drop in repair efficiency in *tid1Δ* mutants (*Piazza et al., 2019*). Here, we further show that specifically during mating-type switch, there is a 20% drop in cell viability post break induction in *tid1Δ* mutants with a 2.2 kbp homologous donor (*Figure 6D*), while no change in viability with shorter 148 nt homology region. Together, these data suggest that in absence of Tid1, long, highly stable D-loops may persist, leading to break-induced toxicity. Additionally, long D-loops are prone to double Holliday junction formation and are more likely to form crossovers. Crossovers could be more deleterious in haploid cells, adding to cytotoxicity, as somatic crossovers are associated with chromosome missegregation (*Chua and Jinks-Robertson, 1991*). Moreover, it may be beneficial to haploid cells to have shorter D-loops and thus reduce the chance of mutagenesis on the single-stranded displaced strand. Together these observations corroborate the haploid-specific physiological importance of Tid1 in promoting repair fidelity and mating type switch.

The ability to Tid1 to restrict D-loop length may also help explain the 70-fold decrease in interchromosomal template switches (ICTC) seen in *tid1Δ* mutants (*Tsaponina and Haber, 2014*). Formation of short, unstable D-loops by Tid1 in haploids may promote template switching via increased D-loop dynamicity. Congruent to our model, the ICTC requires Rad51-binding by Tid1 (*Tsaponina and Haber, 2014*). However, ICTC also requires the Tid1 ATPase activity. In line with

this, *tid1-KR*, like *tid1Δ* also reduces repair efficiency and non-crossover outcomes (*Piazza et al., 2019*), alluding to an additional role for Tid1 downstream in the repair process that requires its ATPase activity. Tid1 translocation on dsDNA also prevents non-recombinogenic binding of Rad51 to dsDNA (*Shah et al., 2010*). The Tid1 ATPase activity is required to turn off Rad53 activation and during checkpoint adaptation (*Ferrari et al., 2013*). Thus, a secondary role of Tid1 and its ATPase activity downstream in mitotic repair remains yet to be delineated. Additionally, despite suppression of *TID1* expression, Tid1 also plays a role in diploid cells, as evident from diploid-specific *tid1Δ* lethality in response to methyl methanesulphonate (MMS) (*Klein, 1997*) and the synthetic lethality of *tid1Δ* with *srs2Δ* seen only in diploids (*Klein, 2001*). These diploid-specific roles may be less dependent on Tid1 concentration relative to Rad54. Nevertheless, our data depicts a direct effect of Tid1 on D-loop formation with subsequent consequences on repair outcome and fidelity in haploid yeast.

# Materials and methods

## Key resources table

| Reagent type (species) or resource | Designation | Source or reference | Identifiers | Additional information |
|---|---|---|---|---|
| Strain, strain background (*Saccharomyces cerevisiae*) | | | | See *Supplementary file 1* |
| Antibody | Tid1 antibody (rabbit polyclonal) | Heyer laboratory | | 1:200 |
| Antibody | GAPDH antibody (mouse monoclonal) | Invitrogen | Catalog: #MA5-15738 | 1:5000 |
| Recombinant DNA reagent | pWDH597 | *Wright and Heyer, 2014* | Amp, URA3 markers | Plasmid for overexpression of *S. cerevisiae* Tid1/Tid1-KR N-terminally tagged with GST, removable by cleavage with PreScission protease. |
| Recombinant DNA reagent | pUC19 | Addgene | Catalog: #50005 | Plasmid used to test topoisomerase contamination of purified protein |
| Recombinant DNA reagent | pBSphix1200 | *Wright and Heyer, 2014* | Amp | Plasmid used as dsDNA donor in D-loop assay in supercoiled or linear form |
| Sequence-based reagent | *100*-mer | *Wright and Heyer, 2014* | | ctggtcataatcatggtggcgaataagta cgcgttcttgcaaatcaccagaaggcggt tcctgaatgaatgggaagccttcaagaa ggtgataagcagga |
| Sequence-based reagent | ds98-*197*-78ss | *Wright and Heyer, 2014* | | Homologous sequence, 197 nt: ctggtcataatcatggtggcgaataagt acgcgttcttgcaaatcaccagaaggc ggttcctgaatgaatgggaagccttca agaaggtgataagcaggagaaacata cgaaggcgcataacgataccactgaccc tcagcaatcttaaacttcttagacgaatca ccagaacggaaaacatccttcatagaaattt |

*Continued on next page*

*Continued*

| Reagent type (species) or resource | Designation | Source or reference | Identifiers | Additional information |
|---|---|---|---|---|
| Sequence-based reagent | ds98-607 | *Wright and Heyer, 2014* | | Homologous sequence, 607 nt: gaagtcatgattgaatcgcgagtggtcggc agattgcgataaacggtcacattaaatttaa cctgactattccactgcaacaactgaacgga ctggaaacactggtcataatcatggtggcga ataagtacgcgttcttgcaaatcaccagaag gcggttcctgaatgaatgggaagccttcaag aaggtgataagcaggagaaacatacgaagg cgcataacgataccactgaccctcagcaatct taaacttcttagacgaatcaccagaacggaa aacatccttcatagaaatttcacgcggcggca agttgccatacaaaacagggtcgccagcaata tcggtataagtcaaagcacctttagcgttaaggt actgaatctctttagtcgcagtaggcggaaaac gaacaagcgcaagagtaaacatagtgccatgc tcaggaacaaagaaacgcggcacagaatgttt ataggtctgttgaacacgaccagaaaactggcc taacgacgtttggtcagttccatcaacatcatagc cagatgcccagagattagagcgcatgacaagta aaggacggttgtcagcgtcataagaggttttac |
| Sequence-based reagent | ds98-931 | Heyer laboratory | | Homologous sequence, 931 nt: gaacggaaaacatccttcatagaaatttcac gcggcggcaagttgccatacaaaacagggt cgccagcaatatcggtataagtcaaagcac ctttagcgttaaggtactgaatctctttagtc gcagtaggcggaaaacgaacaagcgcaa gagtaaacatagtgccatgctcaggaaca aagaaacgcggcacagaatgtttataggtc tgttgaacacgaccagaaaactggcctaac gacgtttggtcagttccatcaacatcatagcc agatgcccagagattagagcgcatgacaag taaaggacggttgtcagcgtcataagaggtt ttacctccaaatgaagaaataacatcatggt aacgctgcatgaagtaatcacgttcttggtc agtatgcaaattagcataagcagcttgcag acccataatgtcaatagatgtggtagaagt cgtcatttggcgagaaagctcagtctcagg aggaagcggagcagtccaaatgtttttgag atggcagcaacggaaaccataacgagcat catcttgattaagctcattagggttagcctcg gtacggtcaggcatccacggcgctttaaaat agttgttatagatattcaaataaccctgaaac aaatgcttagggatttttattggtatcagggtta atcgtgccaagaaaagcggcatggtcaatat aaccagtagtgttaacagtcgggagaggagt ggcattaacaccatccttcatgaacttaatccac tgttcaccataaacgtgacgatgagggacataa aaagtaaaaatgtctacagtagagtcaatagca aggccacgacgcaatggagaaagacggagagc gccaacggcgtccatctcgaaggagtcgccagcg ataaccggagtagttgaaatggtaataagac |

*Continued on next page*

*Continued*

| Reagent type (species) or resource | Designation | Source or reference | Identifiers | Additional information |
|---|---|---|---|---|
| Sequence-based reagent | ds98-915 | Heyer laboratory | | Homologous sequence, 915 nt: gaagtcatgattgaatcgcgagtggtcggcagatt gcgataaacggtcacattaaatttaacctgactatt ccactgcaacaactgaacggactggaaacactgg tcataatcatggtggcgaataagtacgcgttcttgc aaatcaccagaaggcggttcctgaatgaatgggaa gccttcaagaaggtgataagcaggagaaacatac gaaggcgcataacgataccactgaccctcagcaat cttaaacttcttagacgaatcaccagaacggaaaa catccttcatagaaatttcacgcggcggcaagttgc catacaaaacagggtcgccagcaatatcggtata agtcaaagcacctttagcgttaaggtactgaatctc tttagtcgcagtaggcggaaaacgaacaagcgca agagtaaacatagtgccatgctcaggaacaaaga aacgcggcacagaatgttttataggtctgttgaacac gaccagaaaactggcctaacgacgtttggtcagttc catcaacatcatagccagatgcccagagattagag cgcatgacaagtaaaggacggttgtcagcgtcataa gaggttttacctccaaatgaagaaataacatcatggta acgctgcatgaagtaatcacgttcttggtcagtatgca aattagcataagcagcttgcagacccataatgtcaat agatgtggtagaagtcgtcatttggcgagaaagctc agtctcaggaggaagcggagcagtccaaatgtttt tgagatggcagcaacggaaaccataacgagcatc atcttgattaagctcattagggttagcctcggtacg gtcaggcatccacggcgctttaaaatagttgttat agatattcaaataaccctgaaacaaatgc |
| Sequence-based reagent | ds98-915-78ss | Heyer laboratory | | Homologous sequence, 915 nt: gaagtcatgattgaatcgcgagtggtcg gcagattgcgataaacggtcacattaaa tttaacctgactattccactgcaacaactg aacggactggaaacactggtcataatca tggtggcgaataagtacgcgttcttgcaaa tcaccagaaggcggttcctgaatgaatggg aagccttcaagaaggtgataagcaggaga aacatacgaaggcgcataacgataccactg accctcagcaatcttaaacttcttagacgaat caccagaacggaaaacatccttcatagaaa tttcacgcggcggcaagttgccatacaaaa cagggtcgccagcaatatcggtataagtca aagcacctttagcgttaaggtactgaatct ctttagtcgcagtaggcggaaaacgaaca agcgcaagagtaaacatagtgccatgctc aggaacaaagaaacgcggcacagaatg tttataggtctgttgaacacgaccagaaaa ctggcctaacgacgtttggtcagttccatc aacatcatagccagatgcccagagatta gagcgcatgacaagtaaaggacggttg tcagcgtcataagaggttttacctccaaat gaagaaataacatcatggtaacgctgcat gaagtaatcacgttcttggtcagtatgcaa attagcataagcagcttgcagacccataat gtcaatagatgtggtagaagtcgtcatttgg cgagaaagctcagtctcaggaggaagcgga gcagtccaaatgttttttgagatggcagcaacg gaaaccataacgagcatcatcttgattaagct cattagggttagcctcggtacggtcaggcatcc acggcgctttaaaatagttgttatagatatt caaataaccctgaaacaaatgc |
| Sequence-based reagent | HOCSp3; MATp13 | *Mehta et al., 2017* | | Primer pair for measuring extension of D-loops. HOCsp3: GACAAAATGCAGCACGGAAT MATp13: GTTAAGATAAGAACAAAGAAgGATGCT |

*Continued on next page*

*Continued*

| Reagent type (species) or resource | Designation | Source or reference | Identifiers | Additional information |
|---|---|---|---|---|
| Sequence-based reagent | olWDH1760 olWDH1761 | *Piazza et al., 2019* | | Primer pair for reference locus *ARG4* on Ch. VIII. olWDH1760: AGACAGAATTGGCAAAGATCC olWDH1761: GGCCAATTAGTTCACCAAGACG |
| Sequence-based reagent | olWDH1766 olWDH1767 | *Piazza et al., 2019* | | Primer pair for measuring dsDNA integrity at the *HOcs*. olWDH1766: GTTTCAGCTTTCCGCAACAG olWDH1767: GGCGAGGTATTGGATAGTTCC |
| Peptide, recombinant protein | GST-Tid1 | *Nimonkar et al., 2007* | | |
| Peptide, recombinant protein | GST-Tid1-K318R | *Nimonkar et al., 2007* | | |
| Peptide, recombinant protein | Rad54 | *Wright and Heyer, 2014* | | |
| Peptide, recombinant protein | Rad51 | *Van Komen et al., 2006* | | |
| Peptide, recombinant protein | RPA | *Binz et al., 2006* | | |
| Peptide, recombinant protein | *Bsa*1 | New England Biolabs | Catalog: #R0535S | To linearize pBSphix1200 |
| Peptide, recombinant protein | T4 Polynucleotide kinase | New England Biolabs | Catalog: #M0201S | |
| Peptide, recombinant protein | Phusion-U polymerase | Thermo Fischer | Catalog: #PN-F555S | |
| Chemical compound, drug | NADH | Sigma | Catalog: #606-68-6 | ATPase assay |
| Chemical compound, drug | Sera-Mag SpeedBead Carboxylate-Modified Magnetic particles, hydrophobic | Sigma | Catalog: #PN-65152105050250 | DMA |
| Chemical compound, drug | AMPure PB | Pacific Biosciences | Catalog: #100-265-900 | DMA |
| Commercial assay or kit | Baker Flex, Cellulose PEI-F | Fischer Scientific | Catalog: #9004-34-6 | ATPase assay |
| Commercial assay or kit | Epitect Bisulfite kit | Qiagen | Catalog: #59104 | DMA |
| Commercial assay or kit | SMRTbell Template Prep Kit 1.0 | Pacific Biosciences | Catalog: #100-259-100 | DMA |

## Protein purification

GST-Tid1 and its ATPase-defective mutant GST-Tid1-K318R were purified as in *Nimonkar et al., 2012*. The purity of proteins was estimated to be >99% as determined by 12% SDS-PAGE. The concentration of each protein was measured spectrophotometrically using a molar extinction coefficient of 106,800 $M^{-1}cm^{-1}$ at 280 nm. The purified proteins were determined to be free of contaminating ssDNA- and dsDNA-specific nucleases as incubation of an ~20 fold molar excess of either protein over 100-mer ssDNA or 3 kb dsDNA plasmid for 1 hr at 30°C did not generate degradation products. The purified proteins were also devoid of topoisomerase contamination as incubation for 1 hr at 30°C with supercoiled plasmid in presence of ATP did not result in topological changes. Rad51 (*Van Komen et al., 2006*), Rad54 (*Kiianitsa et al., 2002*), and RPA (*Binz et al., 2006*) were purified as described.

## ssDNA substrate production

All ssDNA substrates with different homology lengths to the dsDNA donor, and presence or absence of a non-homologous 3'-flap were created as described in *Wright and Heyer, 2014*. Apart from the substrates described in *Wright and Heyer, 2014*, different long single stranded substrates such as ds98-*931*, ds98-*915*, ds98-*915*-78ss were also created similarly (*Shah et al., 2020*).

## In vitro D-loop assay

In vitro D-loop reactions were performed as described in *Wright and Heyer, 2014* using linear donors or supercoiled plasmid donors as specified. All D-loop reactions were carried out at 30°C. Unless otherwise specified, homologous ssDNA was present at 2.8 µM nt (~3–10 nM molecule depending on the substrate length), donor dsDNA was present at 9 µM nt (3 kb, 3 nM molecules), Rad51 was saturating with respect to the invading ssDNA at 1 Rad51 to 3 nts ssDNA, RPA was at 1 heterotrimer to 25 nt ssDNA, and Rad54 was at 18 nM monomers. If Tid1 was present, it was added 1-, 3-, 7-, or 14- folds over the Rad54 concentration (i.e. 18 nM, 54 nM, 126 nM, or 252 nM). The order of addition was: Rad51/ssDNA, 10 min; then RPA, 5 min; +/- Tid1, 5 min; and finally, Rad54/dsDNA, 15 min. In case of supercoiled dsDNA/Rad54, the reaction was carried on for 10 min, to achieve maximum D-loop formation and prevent D-loop disruption as per *Wright and Heyer, 2014*. The reactions had a final volume of 20 µl. Reactions were stopped with 2 mg/ml Proteinase K (2 µl of 20 mg/ml Proteinase K), 0.2% SDS (0.2 µl of 10% SDS), 10 mM EDTA (0.4 µl of 0.5 M EDTA), and 1x DNA loading dye for gel visualization (2.8 µl of 6x dye). The samples were incubated at room temperature (RT) for 1–2 hr before loading on a 0.8% TBE agarose gel. The electrophoresis was carried out at 70 V for ~3 hr, and the gel was stained with SYBR gold stain for 30 min at RT, before visualization.

### Strand invasion reaction with Tid1 or Rad54 individually

D-loop reactions with Tid1 or Rad54 were also similarly carried out with supercoiled dsDNA and an end-labeled substrate. ds98-*931* was end-labeled with standard PNK procedure and [gamma-$^{32}$P]-ATP. 3 nM molecule end-labeled ds98-*931* substrate was incubated with 1 µM Rad51 for 10 min, then 123 nM RPA for 10 min, and finally 21 nM molecule supercoiled pBSphix1200 (3022 bp dsDNA) along with either Rad54 (100 nM) or Tid1 (100 or 300 nM) for 10 min. The reaction was carried out as described in *Petukhova et al., 2000* at 30°C. The D-loop reactions were stopped as above and separated on a 0.8% agarose gel. The gel was dehydrated, and the radioactively labeled D-loops and substrates were visualized with a STORM 820 phosphorimager.

### D-loop reactions for DMA Assay

D-loop reactions subjected to DMA (see below) were performed similarly, as described in *Shah et al., 2020*. The reaction volume was increased to 25 µl, so that the same D-loop samples can be visualized by gel assay as well as analyzed by DMA assay. DMA was then performed as described below.

## ATP hydrolysis assay

The hydrolysis of ATP in presence of Rad51 was measured using a spectrophotometric assay that coupled production of ADP to the oxidation of NADH. The assay was performed as described previously in *Nimonkar et al., 2012* to test the ATP hydrolysis of purified Tid1.

This ATP hydrolysis assay was also used to test effect of Tid1-KR on Rad54 ATP hydrolysis with the following modifications. 2.2 nM pBSphix1201 (3 kb) dsDNA was incubated with 500 nM Rad51 for 5 min, followed by the addition of 10 nM, 30 nM, or 70 nM Tid1-KR with 5 min incubation and finally, 10 nM Rad54 was added. Alternatively, Rad54 was added to dsDNA and Rad51 containing mix before adding 70 nM Tid1-KR, with similar 5 min incubations between each addition. Tid1-KR was used here instead of Tid1 to distinguish between the ATP hydrolysis contribution from Rad54 and Tid1. As controls, ATPase activity of 70 nM Tid1-KR in presence of Rad51 and dsDNA was tested. We also tested 10 nM Rad54 ATPase activity on dsDNA. The buffer used was similar to the D-loop assay buffer having 35 mM Tris-HCl pH 7.5, 2 mM ATP, 7 mM Mg-acetate, 100 mM NaCl, 0.25 mg/ml BSA, 1 mM TCEP, 5 mM phosphoenolpyruvate, 0.16 mg/ml NADH, 30 U/ml L-Lactate Dehydrogenase (Sigma) and 30 U/ml Phosphocreatine Kinase (Sigma). All reactions were blanked

with the buffer containing dsDNA. NADH conversion factor of 9880 µM min$^{-1}$ was used to calculate ATPase activity as k$_{cat}$ in min$^{-1}$ from the time course.

In absence of Rad51, since Rad54 is expected to have low ATP hydrolysis activity, the activity was measured using a more senstive Thin Layer Chromatography (TLC)-based ATPase assay. Reactions were performed using 3 kb dsDNA (6 µM nt) in a buffer described above. 10 nM Rad54 was added to the reaction 1 min after adding 0, 10, 30, or 70 nM Tid1-KR. Reaction contained 500 µM cold ATP, that was spiked with 0.3 µCi γ$^{P32}$ATP in a 20 µl reaction volume. Samples were collected at 0, 5, 10, 15, 30, and 60 min. The reaction was stopped using 30 mM ATP, 30 mM ADP and 100 mM EDTA before separating on a PEI-Cellulose TLC paper. All reactions were blanked with the buffer containing dsDNA. ATPase activity was calculated as µM min$^{-1}$ from the initial 15 min.

## Yeast strains and genetics

Haploid strains WDHY4999 (*WT MATa*) and WDHY4528 (*tid1 MATα*) from *Piazza et al., 2019* were mated and sporulated to generate WDHY5358 (*tid1 MATa*). Similarly WDHY5511 (*WT MATα*) and WDHY 4704 (*tid1-KR MATa*) were mated and sporulated to generate WDHY5355 (*tid1-KR MATα*). Finally, the opposing types for *WT*, *tid1*, and *tid1-KR* were mated and selected for diploids (*Supplementary file 1*).

Overexpression of Tid1 and Tid1-KR was obtained by transformation of haploid and diploid strains with GST-Tid1 and GST-Tid1-KR expression plasmids created in pWDH597 (*Nimonkar et al., 2007*) with a URA3 selection marker.

Z2216 and Z148 strains for the primer extension assay, survival assay and crossover assay were obtained from *Mehta et al., 2017*. Both strains were transformed with a linear DNA fragment to generate *tid1::URA3*.

## Western blot

Proteins were extracted from 2 × 10$^7$ cells as per standard TCA procedure (*Janke et al., 2010*). Endogenous Tid1 protein was detected by an in-house rabbit anti-Tid1 antibody used at 1:200 dilution, and GAPDH was detected with mouse anti-GAPDH antibody GA1R from Thermo Scientific (MA5-15738, lot QG215126) at a 1:5000 dilution.

## D-loop capture assay

The D-loop capture assay was performed as described in *Piazza et al., 2019*. For strains transformed with URA3 expression plasmid, cells were grown in synthetic SD media devoid of uracil amino acid to maintain the plasmid.

## Non-denaturing single molecule D-loop mapping coupled to PacBio sequencing

In vitro D-loop reactions as described above were performed with a final reaction volume of 25 µl. Reactions were stopped with 2 mg/ml Proteinase K and 10 mM EDTA, before splitting such that 9 µl of the reaction was added to a tube containing 0.2% SDS (0.2 µl of 10% SDS) and 1x DNA loading dye for gel visualization (2.8 µl of 6x dye) as described above. The rest of the 19 µl D-loop reaction was incubated at room temperature (RT) for 30 min to allow deproteinization, before proceeding to bisulfite treatment for the DMA. SDS was avoided in the stop buffer for the DMA fraction so as to prevent branch migration of D-loops (*Allers, 2000*). The 19 µl reaction volume ensures that at least >50 ng of dsDNA was incorporated into the bisulfite reaction.

### D-loop Mapping Assay (DMA)

The D-loops were bisulfite treated using the Qiagen 'Epitect Bisulfite Kit' at room temperature for 3 hr. The bisulfite-treated DNA was PCR amplified using the UNI+Donor-PB-F and UNI+Phix-PB-R primers in the first round, followed by AMPure purification. The amplicons were barcoded in the second round of PCR and AMPure purified. The barcoded samples were pooled in equimolar concentrations, library prepped and subjected to PacBio single molecule real time sequencing on a Sequel-I or II system as described in *Shah et al., 2020*.

## Computational processing

The processing of sequencing reads was as described in *Shah et al., 2020*. The Gargamel pipeline (available at https://github.com/srhartono/footLoop; *Hartono, 2020*) allows user to map reads, assign strands, call single molecule D-loop footprints as peaks of C to T conversion, perform clustering on peaks, and visualize the data. The distribution of D-loop lengths and position were analyzed as in *Shah et al., 2020*.

## Analysis of D-loop levels

The D-loop bands from agarose gel were quantified as percentage of the total dsDNA donor. Similarly, for the DMA, D-loops were quantified as the percentage of top-strand reads containing a footprint. Thus, in this way, for both D-loop quantification methods, D-loops are measured relative to the dsDNA donor. Note that our D-loop levels are not directly comparable to the one in *Wright and Heyer, 2014* due to two reasons. One, in *Wright and Heyer, 2014*, radiolabeled substrates prompted D-loop quantitation in relation to the unused substrate, rather than the donor. Second, D-loop formation can be driven in vitro by excess of one DNA component, dsDNA, when using radiolabeled ssDNA. In most cases, dsDNA was seven times excess over ssDNA (*Wright and Heyer, 2014*). However, we used an almost equimolar ratio of donor and ssDNA to avert superfluous dsDNA from dominating the output in DMA. Finally, the D-loops formed in presence of Tid1 were normalized to the D-loops formed in absence of Tid1 to allow direct comparisons.

## Primer extension assay

The primer extension assay was performed as described in *Mehta et al., 2017*.

## Survival assay

Strains were grown in YEP-Lactate to reach a density of $4 \times 10^6$ cells/ml. Equal number of cells were plated on YEP containing 2% glucose and YEP with 2% galactose. Viability was measured as a ratio of colonies formed on YEP galactose to YEP glucose (*Mehta et al., 2017*).

## Quantitation and statistical analysis

Quantitation of gels was done using ImageJ software. Statistical tests were performed as indicated for each assay.

# Acknowledgements

We thank members of the Heyer laboratory for stimulating discussions. We especially thank Diedre Reitz and Shih-Hsun Hung for providing valuable feedback on the manuscript. We also thank members of the Chedin laboratory, in particular Lionel Sanz for providing AMPure beads and Maika Malig for technical help. We are grateful to Jim Haber for sending strains. We also thank the DNA core technologies at UC Davis and the UC Berkeley Genomic Facility for providing PacBio sequencing services. This research used core services supported by P30 CA93373 and was supported by NIH grants GM58015 and CA92276 to W.-D.H. and NIH grant GM120607 to FC.

# Additional information

## Competing interests

Wolf-Dietrich Heyer: Reviewing editor, *eLife*. The other authors declare that no competing interests exist.

## Funding

| Funder | Grant reference number | Author |
| --- | --- | --- |
| National Institutes of Health | R01GM58015 | Wolf-Dietrich Heyer |
| National Institutes of Health | R01CA92276 | Wolf-Dietrich Heyer |
| National Institutes of Health | P30CA93373 | Wolf-Dietrich Heyer |

| National Institutes of Health | R01GM120607 | Frédéric Chédin |

The funders had no role in study design, data collection and interpretation, or the decision to submit the work for publication.

## Author contributions

Shanaya Shital Shah, Conceptualization, Data curation, Software, Formal analysis, Investigation, Visualization, Writing - original draft, Writing - review and editing; Stella Hartono, Software, Writing - review and editing; Aurèle Piazza, Investigation, Writing - review and editing; Vanessa Som, Investigation; William Wright, Resources, Writing - review and editing; Frédéric Chédin, Software, Supervision, Writing - review and editing; Wolf-Dietrich Heyer, Conceptualization, Formal analysis, Supervision, Funding acquisition, Project administration, Writing - review and editing

## Author ORCIDs

Shanaya Shital Shah (iD) https://orcid.org/0000-0002-2881-2794
Aurèle Piazza (iD) http://orcid.org/0000-0002-7722-0955
Wolf-Dietrich Heyer (iD) https://orcid.org/0000-0002-7774-1953

## Decision letter and Author response

Decision letter https://doi.org/10.7554/eLife.59112.sa1
Author response https://doi.org/10.7554/eLife.59112.sa2

## Additional files

### Supplementary files

• Source data 1. Source data for all the figures.

• Supplementary file 1. *Saccharomyces cerevisiae* strains used in this work. [1] W303 strain background. [2] S288c strain background. [3] Obtained from *Mehta et al., 2017*.

• Transparent reporting form

### Data availability

All data generated or analyzed during this study are included in the manuscript and supporting files. Source data files have been provided for all numerical data.

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
