## [Decision Letter]

**Acceptance summary:**

DNA displacement loops (D-loops) are important intermediates in homologous recombination. The stability of D-loops influences recombination outcomes in meiotic and somatic cells and a number of proteins act on these structures. In this study, Shah and colleagues apply a novel in vitro mapping assay (detailed in the accompanying paper) along with complementary approaches to characterize the role of Rad54 paralog Rdh54 (Tid1) in the formation of D-loops, which are important intermediates in homologous recombination. The authors show that Rdh54 (Tid1) limits the length of D-loops and propose the mechanism for this activity.

**Decision letter after peer review:**

Thank you for submitting your article "Rdh54/Tid1 inhibits Rad51-Rad54-mediated d-loop formation and limits D-loop length" for consideration by *eLife*. Your article has been reviewed by three peer reviewers, and the evaluation has been overseen by Maria Spies as the Reviewing Editor and Jessica Tyler as the Senior Editor. The following individuals involved in review of your submission have agreed to reveal their identity: Anna Malkova (Reviewer #1); Andrew J Deans (Reviewer #2).

The reviewers have discussed the reviews with one another and the Reviewing Editor has drafted this decision to help you prepare a revised submission.

We would like to draw your attention to changes in our revision policy that we have made in response to COVID-19 (https://elifesciences.org/articles/57162). Specifically, when editors judge that a submitted work as a whole belongs in *eLife* but that some conclusions require a modest amount of additional new data, as they may with your paper, we are asking that the manuscript be revised to either limit claims to those supported by data in hand, or to explicitly state that the relevant conclusions require additional supporting data.

Summary:

In this study, Shah and colleagues apply a novel in vitro mapping assay and several complementary approaches to characterize the role of Rad54 paralog Rdh54 (Tid1) in the formation of D-loops, which are important intermediates in homologous recombination. The stability of D-loops influences recombination outcomes in meiotic and somatic cells and a number of proteins have been shown to act on these structures. This manuscript explores in detail the intersection of RAD54 and Rdh54 (Tid1) activity in regulating the D-loop formation and length, and explains most of the complex phenotypic interactions of these two genes in haploid and diploid cells. The authors show that Rdh54 (Tid1) limits the length of D-loops. Complementing the in vitro studies, the authors demonstrate that the level of D-loops formed in vivo, using a previously validated D-loop capture assay, depends on the cellular concentration of Tid1. Overall, the reviewers and the reviewing editor agree that this is an excellent manuscript, its conclusions are well supported by the data, and the results will be of interest to the broad readership of e*Life*, especially to those interested in the mechanisms of genetic recombination, genomic instability and DNA repair.

Essential revisions:

1) The reviewers have a concern with mating type switching experiments, which they feel needs to be addressed. Specifically, the reviewers felt that an explanation of how longer homology can lead to faster repair, and, yet, to more death in the absence of Tid1 is insufficient. Is the explanation the same as provided in your previous work (Piazza et al., 2019), which also reported a decrease in viability in *tid1* relative to WT using longer homology and showed that it is because of loss of non-crossover recombinants possibly due to trapped intermediates that are not resolved? If this is the case and because the *MAT* switching assay in the current manuscript is very similar to that reported before, the inclusion of the data does not add much value to this paper without additional discussion.

2) Another point that needs better explanation is the inference that the presented data in these experiments might help to explain donor preference.

3) The reviewers also have several concerns with respect to interpretation of Tid1-KR mutant data, as this mutant may have poisoning effects by being stuck on DNA. Tid1 is proposed to act as a physical roadblock to Rad54's translocation activity, however this is only really true of the ATPase defective mutant. WT-Tid1 has its own activity, which is lower than that of Rad54. It has not been demonstrated that WT-Tid1 is a roadblock, it probably just acts as a throttle to RAD54. There is a possibility that its activity could be increased by a Rad54 that pushes behind it, is there any evidence for/against this? Roadblock might not be the right word or at least require some clarification in the Abstract. One possible experiment that may clarify the roadblock concept is to use Rad54-KR mutant in combination with WT-Tid1, or WT-Rad54.

4) You suggest that Tid1 competes with Rad54 for binding to Rad51-ssDNA filaments. The evidence for this competition is mostly circumstantial and needs to be acknowledged as such. It seems that an alternative (or additional) explanation for the data is that Tid1 binds to dsDNA, and the stronger inhibition by Tid1-KR could be because of the inability to translocate on dsDNA. Does Tid1-KR inhibit Rad54 ATPase with just a dsDNA substrate (Figure 2—figure supplement 1)?

5) It would be good to have a bit more discussion about a reason for reducing the number/size of D-loops in haploid cells, i.e., is this to favor recombination with a sister chromatid rather than with an unrelated donor molecule, which is less essential in a diploid as the homolog is the most similar unrelated donor?

6) Figure 4 and its discussion are very similar to the counterparts in the accompanying method paper. How the data in this figure differ from that in the method paper? Would it be sufficient to just cite that work when discussing the method, rather than duplicating the data in this manuscript?

---

## [Author Response]

Essential revisions:1) The reviewers have a concern with mating type switching experiments, which they feel needs to be addressed. Specifically, the reviewers felt that an explanation of how longer homology can lead to faster repair, and, yet, to more death in the absence of Tid1 is insufficient. Is the explanation the same as provided in your previous work (Piazza et al., 2019), which also reported a decrease in viability in tid1 relative to WT using longer homology and showed that it is because of loss of non-crossover recombinants possibly due to trapped intermediates that are not resolved? If this is the case and because the MAT switching assay in the current manuscript is very similar to that reported before, the inclusion of the data does not add much value to this paper without additional discussion.

The repair maybe faster, but the efficiency of repair is also decreased in *tid1* mutant cells with long homology. As mentioned in Piazza et al., 2019, it might be due to loss of non-crossover recombinants, and trapped intermediates. However, our data on repair efficiency differs from that published in Piazza et al., 2019, with our data providing information in the context of mating-type switch. The data in Piazza et al., 2019 uses an ectopic system and different homology lengths. We add further explanation in the manuscript regarding these differences in the subsections “In vivo mating type switching regulation by Tid1” and “Physiological Relevance of D-loop modulation by Tid1”.

2) Another point that needs better explanation is the inference that the presented data in these experiments might help to explain donor preference.

During a *MAT-alpha* to *MAT-a* switch requiring an invasion in *HMR*, the *Z*-end has 239 bp homology at *HMR* or 327 bp homology at *HML*. Alternatively, the *WX-*end has 703 bp homology at *HMR* or 2,180 bp homology at *HML.* Hence for a *MAT-alpha* to *MAT-a* switch, it would be essential to invade a locus with shorter homology irrespective of the invading end. However, during a *MAT-a* to *MAT-alpha* switch requiring invasion in *HML*, the *Z*-end may invade *HML* having 327 bp homology or *HMR* with 239 bp homology. Alternatively, the *WX-*end may invade *HMR* having 1,345 bp homology, or *HML* with 1,433 homology. In this case, it is not required to invade a shorter homology region for the mating type switch, and we hypothesize that the *RE* might come into play here. The *RE* would not promote switch in *MAT-alpha* to *MAT-a*, since the *RE* locus is near *HML.* Thus, in this way, the presented data with D-loop regulation may potentially help in the donor preference. We add this explanation in the Discussion section.

Another possibility is that during mating type switch, the D-loop characteristics may determine whether the broken *MAT* molecule invades a potentially unbroken fully homologous sister chromatid or the intrachromosomal *HML*/*HMR* loci. Natural levels of HO-endonuclease may raise the possibility that both sister chromatids may not be cleaved at the same time. However, Klein, 1997, showed that Rdh54 inhibits intra- and inter-chromosomal recombination by ~2-fold. Hence, it seems unlikely that Tid1 promotes intrachromosomal recombination over sister chromatid recombination. This is added to the Discussion.

3) The reviewers also have several concerns with respect to interpretation of Tid1-KR mutant data, as this mutant may have poisoning effects by being stuck on DNA. Tid1 is proposed to act as a physical roadblock to Rad54's translocation activity, however this is only really true of the ATPase defective mutant. WT-Tid1 has its own activity, which is lower than that of Rad54. It has not been demonstrated that WT-Tid1 is a roadblock, it probably just acts as a throttle to RAD54. There is a possibility that its activity could be increased by a Rad54 that pushes behind it, is there any evidence for/against this? Roadblock might not be the right word or at least require some clarification in the Abstract. One possible experiment that may clarify the roadblock concept is to use Rad54-KR mutant in combination with WT-Tid1, or WT-Rad54.

The inhibition on D-loop levels and length is documented in the presence of both WT-Tid1 and Tid1-KR (see Figure 1, Figure 5—figure supplement 3). Hence, it is unlikely that the effect is primarily due to poisoning effects of being stuck on the DNA. We add this explanation in the Results section.

There is no evidence that Tid1 acts as throttle to Rad54, due to another Rad54 pushing behind it. If this were the case, more Rad54 than Tid1 would be required to see a blocking effect, but we see the opposite. More WT Tid1 is required compared to Rad54 to have an effect. Hence, we think that the throttle hypothesis may not be the best explanation.

4) You suggest that Tid1 competes with Rad54 for binding to Rad51-ssDNA filaments. The evidence for this competition is mostly circumstantial and needs to be acknowledged as such. It seems that an alternative (or additional) explanation for the data is that Tid1 binds to dsDNA, and the stronger inhibition by Tid1-KR could be because of the inability to translocate on dsDNA. Does Tid1-KR inhibit Rad54 ATPase with just a dsDNA substrate (Figure 2—figure supplement 1)?

Thank you for this experimental suggestion. We performed an ATPase assay to test if Tid1-KR inhibits the Rad54 ATPase activity on dsDNA in absence of Rad51. We found that the Rad54 ATPase activity decreases only by ~30% with 7x Tid1-KR concentration on dsDNA, which is insignificant compared to the 6-fold drop in activity in presence of Rad51. This result strengthens the argument that Tid1 competes with Rad54 for binding Rad51-ssDNA filaments. We added this data as part E of Figure 2—figure supplement 1. The results are discussed in the subsection “Tid1 competes with Rad54 to inhibit D-loops”.

5) It would be good to have a bit more discussion about a reason for reducing the number/size of D-loops in haploid cells, i.e., is this to favor recombination with a sister chromatid rather than with an unrelated donor molecule, which is less essential in a diploid as the homolog is the most similar unrelated donor?

We think that the one of the reasons for reducing D-loop length/size in haploid cells may be to prevent crossover outcome. If D-loops are longer and more stable/frequent, there is a higher chance of double Holliday Junction formation and subsequently to have a crossover product. A crossover product in a haploid cell, especially during mating-type switch, which may be the most common source of recombination in haploids, would be detrimental to the cells.

To address this, we performed experiments to test the effect of *tid1* on crossover outcome during mating-type switching. We used the same mating-type switch strains with variable homology at Z-end (used for D-loop extension and cell viability test) from Mehta et al., 2016, to measure crossover outcomes by PCR amplification. However, we were unsuccessful in measuring crossover outcome due to technical difficulty in amplifying the modified *HML* locus. So unfortunately, we were unable to successfully complete the experiment.

Another possibility is that during mating type switch, the D-loop characteristics may determine whether the broken *MAT* molecule invades a potentially unbroken fully homologous sister chromatid or the intrachromosomal *HML*/*HMR* loci. Natural levels of HO-endonuclease may raise the possibility that both sister chromatids may not be cleaved at the same time. However, Klein, 1997, showed that Rdh54 inhibits intra- and inter-chromosomal recombination by ~2-fold. Hence, it seems unlikely that Tid1 promotes intrachromosomal recombination over sister chromatid recombination. This is added to the Discussion.

6) Figure 4 and its discussion are very similar to the counterparts in the accompanying method paper. How the data in this figure differ from that in the method paper? Would it be sufficient to just cite that work when discussing the method, rather than duplicating the data in this manuscript?

We show the data to help the reader understand the method with minimal controls without having to consult the accompanying manuscript. We show cumulative data from 3-5 independent replicates, of which only 1-2 overlap with the data reported in the accompanying manuscript. We now mention this in the figure legend. For consistency we only report cumulative data and not averages across the independent replicates in this and the accompanying manuscript.

In addition, we also show the correlation between D-loops quantified from the gel and that from the DMA assay for all D-loop reactions performed in the presence of Tid1/Tid1-KR, which is unique to this manuscript.